# Efficient Randomized Experiments Using Foundation Models

**Piersilvio De Bartolomeis**
ETH Zurich & Harvard University
pdebartol@ethz.ch

**Javier Abad**
ETH Zurich
jabadmartine@ethz.ch

**Guanbo Wang**
Harvard University
gwang@hsph.harvard.edu

**Konstantin Donhauser**
ETH Zurich
donhausk@ethz.ch

**Raymond M Duch**
Oxford University
raymond.duch@ox.ac.uk

**Fanny Yang**
ETH Zurich
fan.yang@inf.ethz.ch

**Issa J Dahabreh**
Harvard University
idahabreh@hsph.harvard.edu

## Abstract

Randomized experiments are the preferred approach for evaluating the effects of interventions, but they are costly and often yield estimates with substantial uncertainty. On the other hand, in silico experiments leveraging foundation models offer a cost-effective alternative that can potentially attain higher statistical precision. However, the benefits of in silico experiments come with a significant risk: statistical inferences are not valid if the models fail to accurately predict experimental responses to interventions. In this paper, we propose a novel approach that integrates the predictions from multiple foundation models with experimental data while preserving valid statistical inference. Our estimator is consistent and asymptotically normal, with asymptotic variance no larger than the *standard* estimator based on experimental data alone. Importantly, these statistical properties hold even when model predictions are arbitrarily biased. Empirical results across several randomized experiments show that our estimator offers substantial precision gains, equivalent to a reduction of up to 20% in the sample size needed to match the same precision as the standard estimator based on experimental data alone.

## 1 Introduction

Randomized experiments are widely considered the preferred approach for evaluating the effects of interventions in scientific research. However, obtaining sufficiently large sample sizes can be costly and time-consuming, especially when studying rare outcomes. For example, Carlisle et al. [11] reported that 481 out of 2579 recently completed clinical trials (19%) failed due to insufficient patient recruitment to meet the required sample size. In cancer trials, this failure rate can be as high as 40% due to strict eligibility and safety requirements [49]. As a result, there is growing interest in exploring in silico experiments as a potential alternative to randomized experiments. In silico experiments leverage the predictions from foundation models [6]—machine learning models trained on massive datasets and applicable to many downstream tasks—to simulate the outcomes of hypothetical randomized experiments. This approach has already shown promising results in replicating the results of randomized experiments in several scientific disciplines, including clinical research [26, 19, 14] and the social sciences [3, 5, 4].

39th Conference on Neural Information Processing Systems (NeurIPS 2025).

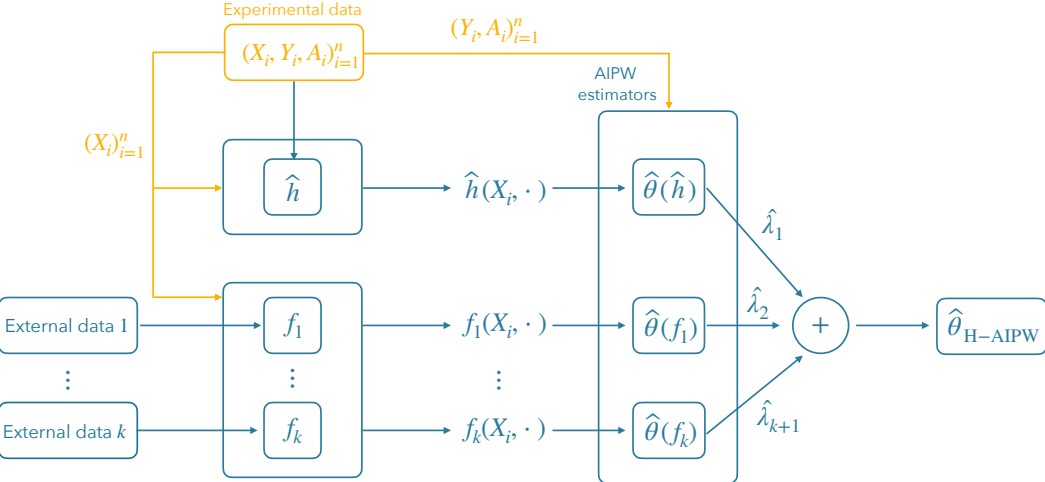

Figure 1: Illustration of the Hybrid Augmented Inverse Probability Weighting (H-AIPW) estimator. For each unit $i$ we observe covariates $X_i$, treatment $A_i$ and outcome $Y_i$; $(X_i, A_i, Y_i)_{i=1}^n$ forms the experimental data. An outcome regression model $\widehat{h}$ fitted to this sample yields the standard AIPW estimate $\widehat{\theta}(\widehat{h})$. In addition, foundation models trained on external data provide the candidate outcome regression models $f_1, \ldots, f_k$, which result in $k$ competing AIPW estimates $\widehat{\theta}(f_1), \ldots, \widehat{\theta}(f_k)$. By integrating the outcome regression models trained on a large external sample, rather than fitting a single model on the small experimental sample, H-AIPW can reduce the variance of the average treatment effect estimate.

However, for a method to be adopted in safety-critical fields like medicine, valid statistical inference is an absolute requirement. For instance, the Food and Drug Administration guidance strongly recommends that any method aimed at improving the efficiency of randomized experiments should provide valid inference under minimal statistical assumptions [23]. Yet, statistical inference from in silico experiments is not valid if model predictions fail to reflect experimental responses to interventions. Since such an assumption is difficult to falsify, the growing consensus among researchers is that results from in silico experiments should be limited to exploratory stages of research, for example, pilot studies to predict effect sizes in larger experiments [27].

This limitation raises an important question: Can we safely leverage the predictions from foundation models to improve efficiency while preserving valid statistical inference? In this paper, we introduce the **H**ybrid **A**ugmented **I**nverse **P**robability **W**eighting (H-AIPW), a novel estimator that can integrate predictions from multiple, potentially biased, foundation models while preserving valid statistical inference under minimal assumptions. Specifically, we prove

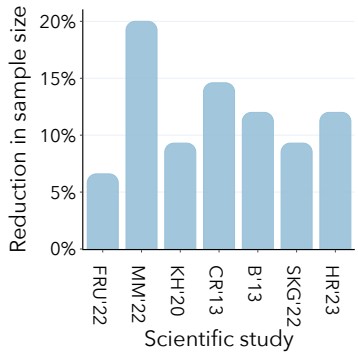

Figure 2: Our estimator achieves the same statistical precision as the standard estimator with up to 20% fewer samples. Each study is subsampled to $n = 75$. We plot the percentage reduction in the sample size needed to match the confidence interval width of the standard estimator using ours.

that H-AIPW is consistent and asymptotically normal, with asymptotic variance no larger than the standard estimator based on experimental data alone. Importantly, our results require no additional assumptions beyond those necessary for estimating treatment effects in classical randomized experiments. While our methodology applies broadly, we focus our empirical results on social science survey experiments, where large language models (LLMs) can provide rich predictive signals. Across several randomized experiments, we show that H-AIPW can offer substantial precision gains, equivalent to a reduction of up to 20% in the sample size required to achieve the same precision as the standard estimator based on experimental data alone (see Figure 2).

## 2 Related work

Our work draws heavily from the literature on semiparametric inference and double machine learning [42, 41, 47, 12]. In particular, our estimator is an optimal combination of several Augmented Inverse Probability Weighting (AIPW) estimators, whose outcome regression models are replaced with foundation models. Importantly, the standard AIPW estimator, which relies on an outcome regression model estimated using experimental data alone, is also included in the combination.

**Integrating foundation models** Prediction-powered inference (PPI) [1] is a statistical framework that constructs valid confidence intervals using a small labeled dataset and a large unlabeled dataset imputed by a foundation model. PPI has been applied in various domains, including generalization of causal inferences [18, 9], large language model evaluation [24, 20], and improving the efficiency of social science experiments [8, 21]. Recent work by Poulet et al. [40] introduces Prediction-powered inference for clinical trials (PPCT), an adaptation of PPI to estimate average treatment effects in randomized experiments without any additional unlabeled data. PPCT combines the difference in means estimator with an AIPW estimator that uses the predictions from one foundation model for both treatment and control groups. However, our work differs in a crucial aspect: PPCT does not include the standard AIPW estimator with the outcome regression model estimated from experimental data. Therefore, there is no mechanism to prevent PPCT from having a higher variance compared to the standard AIPW estimator that uses experimental data alone (see e.g. Table 1). This risk of increased variance is a critical limitation in many settings—for example, in clinical trials, pharmaceutical sponsors are highly risk-averse and methods that carry even a small chance of underperforming the established standard face significant barriers to adoption. We refer the reader to Appendix A.2 for a more complete discussion of the differences between our approach and PPI.

**Integrating observational data** There is growing interest in augmenting randomized experiments with data from observational studies to improve statistical precision [36]. One approach involves first testing whether the observational data is compatible with the experimental data [13, 37, 29, 17, 16], and then combining the datasets to improve precision, if the test does not reject. These tests, however, have low statistical power, especially when the experimental sample size is small, which is precisely when leveraging observational data would be most beneficial. Another line of work combines a biased (but more precise) estimator from observational data with an unbiased estimator from experimental data to obtain a debiased estimate [31, 15, 43, 48]. However, in small sample settings, the debiasing procedure often fails. Closest to ours, there are two lines of works that propose unbiased estimators: one integrates a prognostic score estimated from observational data as a covariate when estimating the outcome regression model [44, 35], while the other incorporates an outcome regression model estimated from observational data directly into the AIPW estimator [25, 32]. However, both approaches rely on access to well-structured observational data to improve statistical precision. In contrast, our approach is not constrained by the availability of well-structured data, and instead leverages black-box foundation models trained on external data sources.

## 3 Background on randomized experiments

We observe a dataset $\mathcal{D}$ of size $n$ from a randomized experiment, containing tuples $(X, Y, A)$ of covariates $X \in \mathbb{R}^d$, bounded outcome $Y \in \mathbb{R}$, and treatment variable $A \in \{0, 1\}$. We assume that the data is drawn i.i.d. from a joint distribution $\mathbb{P}$ over $(X, Y(0), Y(1), Y, A)$, where the potential outcomes $(Y(0), Y(1)) \in \mathbb{R}^2$ are unobserved and $Y = Y(A)$. Our goal is to use $\mathcal{D}$ to estimate the average treatment effect (ATE) in the randomized experiment population,

$$\theta := \mathbb{E}[Y(1) - Y(0)],$$

where the expectation is taken over $\mathbb{P}$. In particular, we want to improve upon the statistical precision of classical ATE estimators by constructing an asymptotically valid confidence interval that is narrower. We further assume that the data is collected from a proper randomized experiment that satisfies the following standard assumptions.

**Assumption 3.1** (Identification assumptions). *The data-generating process satisfies*

$$(i) \ Y(a) \perp\!\!\!\perp A, \text{ for } a = 0, 1.$$
$$(ii) \ \mathbb{P}(A = a \mid X) = \pi_a \in (0, 1), \text{ for } a = 0, 1.$$

We assume that the propensity score $\pi_a$ is known by design, as is the case in the vast majority of experiments. Nevertheless, our framework can be extended to allow for covariate-adaptive randomization or settings where the probability of treatment needs to be estimated.

Under Assumption 3.1, we can identify the ATE as follows

$$\theta = \mathbb{E}[Y(1) - Y(0)] = \mathbb{E}[Y \mid A = 1] - \mathbb{E}[Y \mid A = 0].$$

Therefore, the standard approach is to estimate $\theta$ using the difference in means (DM) estimator,

$$\widehat{\theta}_{\text{DM}} := \frac{1}{n_1} \sum_{i:A_i=1} Y_i - \frac{1}{n_0} \sum_{i:A_i=0} Y_i, \text{ where } n_a = |\{i : A_i = a\}|.$$

This estimator is consistent and asymptotically normal (see e.g. Wager [50, Theorem 1.2]):

$$\sqrt{n}(\widehat{\theta}_{\text{DM}} - \theta) \rightsquigarrow \mathcal{N}(0, V_{\text{DM}}),$$

where $\rightsquigarrow$ denotes convergence in distribution and $V_{\text{DM}}$ is the asymptotic variance. Therefore, provided that we can obtain a consistent estimator of the asymptotic variance, $\widehat{V}_{\text{DM}} = V_{\text{DM}} + o_{\mathbb{P}}(1)$, we can construct an asymptotically valid confidence interval

$$\mathcal{C}_{\text{DM}}^\alpha = \left( \widehat{\theta}_{\text{DM}} \pm z_{1-\frac{\alpha}{2}} \sqrt{\frac{\widehat{V}_{\text{DM}}}{n}} \right), \tag{1}$$

such that $\lim_{n\to\infty} \mathbb{P}(\theta \in \mathcal{C}_{\text{DM}}^\alpha) \geq 1 - \alpha$, where $z_\alpha$ is the $\alpha$-quantile of the standard normal distribution. Arguably, $\widehat{\theta}_{\text{DM}}$ is all that is needed to estimate average treatment effects in randomized experiments. However, the variance $\widehat{V}_{\text{DM}}$ is often very large, leading to a wide confidence interval $\mathcal{C}_{\text{DM}}^\alpha$. In the next section, we will show that it is possible to obtain narrower confidence intervals if we leverage the information contained in the covariates.

## 3.1 A class of valid estimators: Augmented Inverse Probability Weighting

Robins et al. [42] show that, when the propensity score is known, every regular and asymptotically linear estimator of $\theta$ is asymptotically equivalent to an AIPW estimator of the form below:

$$\widehat{\theta}_{\text{AIPW}}(h) := \frac{1}{n} \sum_{i \in \mathcal{D}} \psi_i(h),$$

where $h : \mathbb{R}^d \times \{0, 1\} \to \mathbb{R}$ is a square-integrable function, and

$$\psi_i(h) := \left( \frac{A_i}{\pi_1}(Y_i - h(X_i, 1)) + h(X_i, 1) \right) - \left( \frac{1 - A_i}{\pi_0}(Y_i - h(X_i, 0)) + h(X_i, 0) \right).$$

The most efficient estimator within this class uses an outcome regression model $h$ that minimizes the asymptotic variance. Specifically, the semiparametric efficiency lower bound is attained by choosing $h^\star(x, a) = \mathbb{E}[Y|X = x, A = a]$, which corresponds to the conditional mean of the outcome. In other words, the estimator $\widehat{\theta}_{\text{AIPW}}(h^\star)$ attains the smallest asymptotic variance among all consistent and asymptotically normal estimators of $\theta$, and, thus, the smallest possible confidence interval in large samples. In practice, however, we only have an estimator $\widehat{h}$ of the conditional mean $h^\star$, which achieves the efficiency lower bound only if $||\widehat{h} - h^\star||_{L_2(\mathbb{P})} = o_{\mathbb{P}}(1)$.

Below, we adapt the standard result that establishes consistency and asymptotic normality of the AIPW estimator to our setting, where the treatment probability is known. The key distinction from the standard setting is that asymptotic normality is achieved as long as the outcome regression model has an asymptotic limit. This implies that the confidence intervals are valid even when the outcome regression is estimated using complex machine learning models with unknown convergence rates.

**Proposition 1** (Asymptotic behavior of AIPW). *Let $\tilde{\mathcal{D}}$ be an auxiliary sample, independent of $\mathcal{D}$. Let $\widehat{h}$ be a model trained on $\tilde{\mathcal{D}}$, and let $h^\dagger$ be a square-integrable limit such that for $a \in \{0,1\}$,*

$$\left\|\widehat{h}(\cdot, a) - h^\dagger(\cdot, a)\right\|_{L^2(\mathbb{P})} \xrightarrow{\mathbb{P}^*} 0,$$

*where $\mathbb{P}^*$ denotes the joint law of $(\mathcal{D}, \tilde{\mathcal{D}})$. Then, it follows that $\widehat{\theta}_{\text{AIPW}}(\widehat{h})$ is asymptotically normal:*

$$\sqrt{n}(\widehat{\theta}_{\text{AIPW}}(\widehat{h}) - \theta) \rightsquigarrow \mathcal{N}(0, V_{h^\dagger}),$$

*where $V_{h^\dagger} = \mathbb{E}\left[\left(\psi(h^\dagger) - \theta\right)^2\right]$ is the asymptotic variance.*

We provide a proof of this result in Appendix A.1.1. Proposition 1 shows that the choice of estimator for the outcome regression does not affect the validity of the inference, provided that it is trained on an independent sample—for example, by using cross-fitting on the experimental sample or training the model on a larger external dataset. Under these conditions, we can then construct an asymptotically valid confidence interval $\mathcal{C}^\alpha_{\text{AIPW}}$ as outlined in Equation (1). Further, the asymptotic variance critically depends on the limiting model $h^\dagger$, and decreases as $h^\dagger$ more closely approximates the true conditional mean $h^\star$ (see Appendix A.3 for a formal result on the dependency of the excess variance on the difference between the outcome regression model and $h^\star$).

A standard way to obtain an estimate $\widehat{h}$ using the observed data would be to output the minimizers of the empirical risks of each $a$:

$$\widehat{h}(X, a) \in \underset{h \in \mathcal{H}}{\arg\min} \frac{1}{n_a} \sum_{i:A_i=a} \mathcal{L}(Y_i, h(X_i)), \tag{2}$$

where $\mathcal{H}$ is a chosen model class (e.g. all linear functions) and $\mathcal{L}$ a point-wise loss function (e.g. mean squared loss). We refer to the empirical risk minimizer $\widehat{\theta}_{\text{AIPW}}(\widehat{h})$ in Equation (2) with $\mathcal{H}$ being the linear function class, as the *standard* AIPW estimator—as the name suggests, this is the most common estimator that is currently being deployed in practice. Hence, a key desideratum for any new estimator is *safety* with respect to this standard baseline—that is, it should never perform substantially worse in terms of variance, and ideally perform better than the standard AIPW estimator. However, a key limitation of the standard AIPW estimator is that its outcome regression model is trained on a small sample size and is limited to a simple function class. In the next section, we introduce a novel estimator that instead leverages predictions from foundation models trained on vast amount of external data, significantly improving our chances of learning an accurate outcome regression model.

## 4 Methodology

We introduce **H**ybrid **A**ugmented **I**nverse **P**robability **W**eighting (H-AIPW), an estimator that, in contrast to the standard AIPW, leverages the predictions from multiple foundation models to improve statistical precision. In what follows, we first provide a formal definition of the H-AIPW estimator in Algorithm 1 and then give theoretical results for its asymptotic distribution and variance.

### 4.1 Hybrid Augmented Inverse Probability Weighting

With the recent widespread availability of foundation models, we can potentially improve the accuracy of the outcome regression model beyond what is obtained from Equation (2) simply by replacing it with a foundation model. This is the principle behind PPI-style estimators, yet such an approach offers no safety guarantee of doing no worse than the standard estimator—a critical desideratum for adoption in many domains. Further, as is often the case with language models, multiple competing models may be available, with no clear way to determine the best choice for a given task in advance. Therefore, we propose combining multiple AIPW estimators, each using a different outcome regression model:

$$\widehat{\theta}_{\text{AIPW}}(\widehat{h}), \widehat{\theta}_{\text{AIPW}}(f_1), \ldots, \widehat{\theta}_{\text{AIPW}}(f_k).$$

Here, $\widehat{h}$ is estimated exclusively from experimental data, as shown in Equation (2), while $f_1, \ldots, f_k$ are foundation models trained on independent external data. The challenge of selecting an optimal

---

**Algorithm 1** **H**ybrid **A**ugmented **I**nverse **P**robability **W**eighting (H-AIPW)

---

**Require:** (i) Dataset $\mathcal{D} = \{(X_i, A_i, Y_i)\}_{i=1}^n$. (ii) Collection of foundation models $f_1, \ldots, f_k$.
(iii) Loss function $\mathcal{L}$ and function class $\mathcal{H}$. (iv) $\pi_a$ for $a = 0, 1$. (v) Significance level $\alpha$.

1: Use cross-fitting to compute the estimate $\widehat{\theta}_{\text{AIPW}}(\widehat{h})$ from the dataset $\mathcal{D}$, where for each arm $a$:

$$\widehat{h}(X, a) \in \arg\min_{h \in \mathcal{H}} \left\{ \frac{1}{n_a} \sum_{i:A_i=a} \mathcal{L}(Y_i, h(X_i)) \right\}.$$

2: Compute $\widehat{\lambda} = \widehat{\Sigma}^{-1} \mathbf{1} / (\mathbf{1}^\top \widehat{\Sigma}^{-1} \mathbf{1})$, where

$$\bar{\psi} := \frac{1}{n} \sum_{i=1}^n (\psi_i(\widehat{h}), \ldots, \psi_i(f_k)),$$

$$\widehat{\Sigma} := \frac{1}{n-1} \sum_{i=1}^n \left( (\psi_i(\widehat{h}), \ldots, \psi_i(f_k)) - \bar{\psi} \right)^\top \left( (\psi_i(\widehat{h}), \ldots, \psi_i(f_k)) - \bar{\psi} \right).$$

3: Compute the estimate and its variance

$$\widehat{\theta}_{\widehat{\lambda}} := \widehat{\lambda}_1 \widehat{\theta}_{\text{AIPW}}(\widehat{h}) + \sum_{j=1}^k \widehat{\theta}_{\text{AIPW}}(f_j) \, \widehat{\lambda}_{j+1}, \;\; \text{and} \;\; \widehat{V}_{\widehat{\lambda}} := \widehat{\lambda}^\top \widehat{\Sigma} \, \widehat{\lambda}. \tag{3}$$

4: **Return:** $\mathcal{C}^\alpha_{\text{H-AIPW}} = \left( \widehat{\theta}_{\widehat{\lambda}} \pm z_{1-\frac{\alpha}{2}} \sqrt{\frac{\widehat{V}_{\widehat{\lambda}}}{n}} \right)$, where $z_\alpha$ is the $\alpha$-quantile of the standard normal.

---

estimator from a set of competing estimators for the same target quantity has been extensively studied in the statistical literature; see e.g. Lavancier and Rochet [34]. A common solution is to consider a weighted average of the available estimators, which in our setting corresponds to

$$\widehat{\theta}_\lambda := \lambda_1 \widehat{\theta}_{\text{AIPW}}(\widehat{h}) + \sum_{j=1}^k \widehat{\theta}_{\text{AIPW}}(f_j) \lambda_{j+1}, \; \text{for some } \lambda \in \Lambda = \{\lambda \in \mathbb{R}^{k+1} : \sum_{j=1}^{k+1} \lambda_j = 1\}.$$

We illustrate the estimation pipeline in Figure 1. Further, we restrict the weights to the constraint set $\Lambda$ so that the combined estimator $\widehat{\theta}_\lambda$ is still in the class of AIPW estimators. We can then choose the weights that minimize the asymptotic variance $V_\lambda$ of the combined estimator $\widehat{\theta}_\lambda$, that is:

$$\lambda^\star = \arg\min_{\lambda \in \Lambda} V_\lambda = \arg\min_{\lambda \in \Lambda} \lambda^\top \Sigma \lambda = \Sigma^{-1} \mathbf{1} / (\mathbf{1}^\top \Sigma^{-1} \mathbf{1}),$$

where $\Sigma := \text{Cov}[(\psi(h^\dagger), \ldots, \psi(f_k))^\top]$ is the asymptotic covariance and $h^\dagger$ is the asymptotic limit of $\widehat{h}$. However, in practice, we only have an estimate $\widehat{\Sigma}$ of the covariance matrix, and thus we use

$$\widehat{\lambda} := \arg\min_{\lambda \in \Lambda} \lambda^\top \widehat{\Sigma} \lambda.$$

**Asymptotic validity and efficiency** We now establish that the H-AIPW estimator is consistent and asymptotically normal, with an asymptotic variance that is no greater than that of the standard AIPW.

**Theorem 2** (Asymptotic behavior of H-AIPW). *Let $\widehat{h}$ be an outcome regression model that satisfies the conditions in Proposition 1, with asymptotic limit $h^\dagger$. Further, let $\widehat{\theta}_{\widehat{\lambda}}$ be as in Equation (3), and assume that $\Sigma$ is non-singular and $\|\widehat{\Sigma} - \Sigma\|_{\text{op}} \xrightarrow{p} 0$. Then, it holds that*

$$\sqrt{n}(\widehat{\theta}_{\widehat{\lambda}} - \theta) \rightsquigarrow \mathcal{N}(0, V_{\lambda^\star}).$$

*Moreover, the asymptotic variance of the combined estimator is no greater than that of any individual estimator, i.e. it holds that*

$$V_{\lambda^\star} \leq \Sigma_{jj}, \text{ for } j = 1, \ldots, k+1.$$

<table>
<tr><td>

**Example System Prompt**

You are a 35-year-old female, politically Democrat, holding liberal views. Additionally, your religion is Christianity, and you once or twice a month attend religious services. You reside in a building with two or more apartments, and your household has a yearly income of $85,000 to $99,999.

You are responding to a scenario reflecting a debate involving college campus events and broader social issues.

</td><td>

**Example User Prompt**

**Treatment:** *A student organization denied Antifa's request for a rally, citing safety concerns due to altercations at similar events.*
**Outcome question:** Do you agree or disagree with the statement:
*"Cancel culture is a big problem in today's society."* Choose an integer between 1 (strongly agree) and 5 (strongly disagree).
**Instruction**: Reflect on the scenario and use your reasoning to assign a value.

</td></tr>
<tr><td align="center">(a)</td><td align="center">(b)</td></tr>
</table>

Figure 3: Examples of a system and user prompts used to generate synthetic responses for Fahey et al. [22].

We provide a proof of this result in Appendix A.1.2. Theorem 2 offers a principled approach to combining multiple competing AIPW estimators, ensuring that the resulting estimator is at least as precise (asymptotically) as the best estimator in the ensemble. In particular, this approach allows us to leverage the strengths of foundation models without any risks: when these models give accurate outcome predictions, the combined estimator uses their extra information to improve precision. On the other hand, when the foundation models are biased, the final estimator falls back to AIPW.

## 4.2 Step-by-step recipe with Large Language Models

In this section, we provide a step-by-step guide for practitioners to implement H-AIPW using Large Language Models (LLMs). Our guide focuses on LLMs as they are both widely accessible and have demonstrated good accuracy in predicting human behavior [27]. As a concrete example, we present a political science experiment that evaluates the effect of free speech framings on opposition to cancel culture among Americans [22]. We provide simplified prompts here and refer readers to Appendix C.2 for the full LLM prompts.

1. **Extract participant information.** Extract the tuples $Z_i = (X_i, Y_i, A_i)$ for each participant $i$ in the study. In Fahey et al. [22], covariates include age, gender, ideology, income, and religion. The treatment represents a scenario where an Antifa protest is banned: for safety reasons only ($A = 0$), or for safety reasons and cancel culture ($A = 1$). The outcome is measured on a scale from 1 to 5, as the level of agreement with the statement: *"Cancel culture is a big problem in today's society."*

2. **Construct system prompts.** For each participant $i$, create a *persona* that matches $X_i$ and guides the LLM in simulating responses. In this study, personas summarize the participant's demographics. The persona is then used as the *system* prompt for the LLM (see Figure 3a).

3. **Construct user prompts.** The *user* prompt includes the experimental treatment, the outcome question, and instructions to guide the LLM (see Figure 3b). We prompt the LLM to generate a synthetic outcome for both treatment and control. The final instruction is sampled from a predefined pool to introduce variability in the LLM's responses (see Appendix C.2.9).

4. **Simulate outcome responses.** Query the LLM using the user and system prompts. Validate that the responses are numeric and conform to the specified outcome scale. For experiments where multiple instructions are sampled, compute the average response.

5. **Estimate treatment effects.** Compute the confidence interval $\mathcal{C}^{\alpha}_{\text{H-AIPW}}$ via Algorithm 1. Using cross-fitting to fit the outcome models is key for coverage in small-sample settings.

Table 1: Performance comparison of H-Aipw against baseline estimators (Ppct, Dm, Aipw, Procova) across several randomized experiments. We randomly subsample each study at sample sizes $n = 100$ and $n = 200$. We report the variance of each estimator averaged over $R = 10k$ subsampling repetitions. Cells shaded in blue denote the standard Aipw baseline that should be improved upon using external data; green indicates better precision than standard Aipw; and red indicates worse precision than standard Aipw.

| | Melin et al. (2022) | | Silverman et al. (2022) | | Kennedy et al. (2020) | | Fahey et al. (2023) | |
| **Estimator** | $n = 100$ | $n = 200$ | $n = 100$ | $n = 200$ | $n = 100$ | $n = 200$ | $n = 100$ | $n = 200$ |
|---|---|---|---|---|---|---|---|---|
| H-Aipw | **10.39** | **10.28** | **2.10** | **2.14** | **17.09** | **17.47** | **4.87** | 4.94 |
| Ppct | 11.00 | 11.06 | 2.25 | 2.26 | 17.87 | 17.97 | 4.88 | **4.91** |
| Procova | 11.81 | 10.62 | 2.24 | 2.22 | 18.38 | 18.11 | 5.18 | 5.09 |
| Aipw (boosting) | 12.82 | 12.44 | 2.82 | 2.83 | 23.09 | 23.12 | 6.31 | 6.37 |
| Aipw (standard) | 11.72 | 10.57 | 2.22 | 2.20 | 18.09 | 17.95 | 5.09 | 5.04 |
| Dm | 11.10 | 11.10 | 2.30 | 2.30 | 18.07 | 18.08 | 5.61 | 5.62 |

| | Caprariello et al. (2013) | | Brandt (2013) | | Haaland et al. (2023) | | Shuman et al. (2024) | |
| **Estimator** | $n = 100$ | $n = 200$ | $n = 100$ | $n = 200$ | $n = 100$ | $n = 200$ | $n = 100$ | $n = 200$ |
|---|---|---|---|---|---|---|---|---|
| H-Aipw | **5.88** | **5.96** | **11.86** | **11.90** | **4.49** | **4.44** | **8.46** | **8.91** |
| Ppct | 5.99 | 6.01 | 12.07 | 12.12 | 4.50 | 4.52 | 9.08 | 9.14 |
| Procova | 6.41 | 6.13 | 12.77 | 12.25 | 4.73 | **4.44** | 9.12 | 9.55 |
| Aipw (boosting) | 7.79 | 7.60 | 15.20 | 14.70 | 5.39 | 5.22 | 10.53 | 10.67 |
| Aipw (standard) | 6.39 | 6.18 | 12.55 | 12.13 | 4.82 | 4.55 | 9.20 | 10.31 |
| Dm | 6.15 | 6.15 | 12.81 | 12.80 | 5.72 | 5.71 | 13.83 | 13.83 |

## 5 Experiments

In this section, we first show that H-Aipw improves statistical precision across eight randomized experiments without compromising empirical coverage. We then evaluate the performance of several LLMs, highlighting the importance of both model scale and inference-time compute: larger models (e.g., GPT-4o and LLaMA 3 70B) consistently outperform smaller ones in prediction accuracy, and averaging over multiple prompts at inference time further improves performance.

### 5.1 H-AIPW offers improved precision

We evaluate H-Aipw across eight randomized experiments in Economics [28], Psychology [7], Political Science [22], Foreign Policy [46], Sociology [33, 39, 10, 45]. These studies were selected from the multidisciplinary Time-Sharing Experiments in the Social Sciences (TESS) repository, along the lines of Ashokkumar et al. [4]. For each experimental study $s$, we implement the following subsampling procedure: starting with a full dataset $\mathcal{D}$ of size $N_s$, we select a target sample size $n$. For each subsampling repetition $r \in \{1, \dots, R\}$, we sample $n$ participants without replacement from $\mathcal{D}$, ensuring the treatment and control groups are balanced, to create a smaller dataset $\mathcal{D}_r$.

**Estimators and metrics** We implement H-Aipw by integrating predictions from three LLMs: GPT-4o, Claude 3.5 Haiku, and LLaMA 3 70B. For each LLM, we use 10 different prompts for prediction and average over the responses (see Appendix C.2 for example prompts). We benchmark our estimator against two standard estimators: $\widehat{\theta}_{\mathrm{DM}}$ (Dm) and $\widehat{\theta}_{\mathrm{AIPW}}(\widehat{h})$ (Aipw), where $\widehat{h}$ is the solution to the optimization problem in Equation (2) with either a linear (standard) or complex (boosting) function class. We also implement the concurrent Ppct estimator [40] and the Procova estimator [35, 44], both using GPT-4o as the external model. These two serve as a more competitive baseline that also leverages predictions from foundation models (see Appendix C.1 for implementation details). To benchmark statistical precision, for each estimator $\widehat{\theta}$, we compute the scaled variance $\frac{1}{R} \sum_{r=1}^{R} n \widehat{\mathrm{Var}}[\widehat{\theta}_r]$, where $\widehat{\mathrm{Var}}$ is the empirical variance obtained from the dataset $\mathcal{D}_r$—as the sample size grows, the scaled variance approaches the asymptotic variance of the estimator.

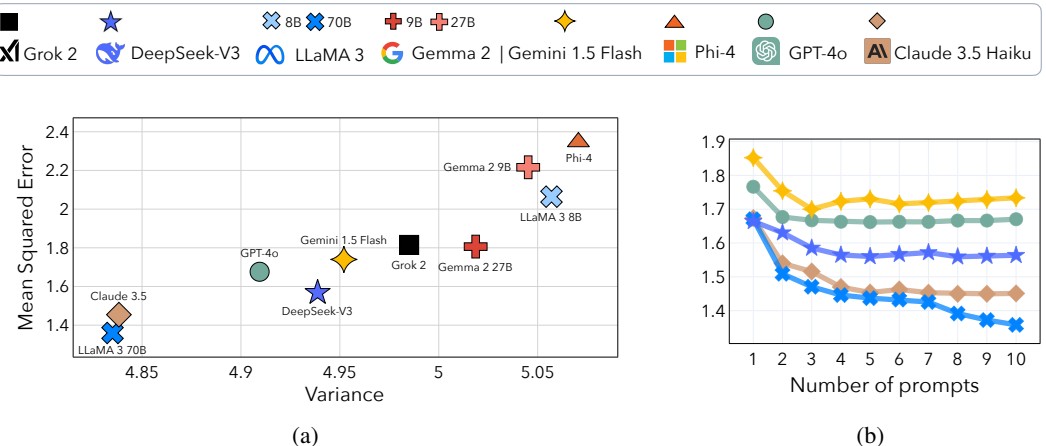

Figure 4: Impact of model scale and inference-time compute on the performance of H-Aipw in the study by Fahey et al. [22]. **(Left) Model scale**: Figure 4a shows the relationship between the estimate of the H-Aipw variance (average on $R = 10k$ repetitions, sample size $n = 50$) and mean squared error (MSE) for LLMs of varying sizes (10 prompts at inference time). **(Right) Inference-time compute**: Figure 4b shows the impact on the MSE of increasing the number of prompts at inference time and averaging the resulting predictions.

**Results**   Table 1 reports the estimated variance of several competing estimators across eight different experimental studies and two sample sizes ($n = 100$ and $n = 200$). Across nearly all scenarios, H-Aipw consistently achieves the lowest variance among all estimators, and hence the tightest confidence interval. In particular, we observe variance reductions of roughly 5–11% compared to the standard Aipw estimator based on experimental data only. The gains are especially pronounced in the small-sample setting ($n = 100$), where reducing variance is most critical. As expected, we observe that the Ppct estimator can be less precise than the standard Aipw estimator. This can be explained by noting that Ppct is only guaranteed (asymptotically) to be at least as precise as the difference in means estimator (Dm). Further, we observe that the Aipw estimator using a complex function class (boosting) suffers from very high variance, as the small sample sizes in the randomized experiments do not allow complex modeling choices. Lastly, while Theorem 2 ensures that H-AIPW provides valid confidence intervals asymptotically, empirical results in Appendix B.1 confirm that all evaluated estimators—including H-Aipw—maintain near-nominal coverage levels in finite samples.

**Image treatments and contamination**   The study by Shuman et al. [45] is particularly relevant for two reasons. First, the data were published in December 2024, after the last known training cutoff for GPT-4o, ensuring it was not included in the model's training set. Second, the treatment is an image rather than text, allowing us to evaluate our statistical framework beyond the text modality. Since the other foundation models do not support image inputs, we rely only on GPT-4o for outcome predictions in this study. Even so, H-Aipw achieves the lowest variance among all baselines, outperforming both Ppct and Procova, suggesting that its gains over other approaches that integrate external models are not only due to the access to multiple models.

## 5.2   Improving the accuracy of LLMs

We now study how two strategies for improving the accuracy of LLMs—model scale and inference-time compute—affect the precision of the H-Aipw estimator. In our setting, increasing inference-time compute boils down to presenting slight variations of the same prompt at inference time and averaging over the responses. We provide the complete list of the prompts used in Appendix C.2.9. For each LLM $f$, we evaluate prediction performance using the Mean Squared Error (MSE) on the full dataset: $\frac{1}{N} \sum_{i=1}^{N} (f(X_i, A_i) - Y_i)^2$. Our findings indicate that larger models and increased inference-time compute can improve prediction accuracy, which in turn can reduce the variance of H-Aipw.

**Model scale** Figure 4a illustrates the precision gains achieved by H-AIPW when leveraging predictions from LLMs of varying scales. We study the relationship between MSE and the estimate of the H-AIPW variance when integrating predictions from *small* models (LLaMA 3 8B, Gemma 2 9B, Phi-4, Gemma 2 27B) and *large* models (LLaMA 3 70B, GPT-4o, Gemini 1.5 Flash, DeepSeek-V3, Claude 3.5 Haiku, Grok 2). Large models consistently achieve lower MSE and thus lower variance than smaller models, with LLaMA 3 70B excelling despite having fewer parameters than GPT-4o and Claude 3.5 Haiku.

**Inference-time compute** Figure 4b shows that averaging over many prompts consistently reduces the MSE for the large models—a similar trend is expected for the smaller ones. As smaller MSE is associated with higher precision (see Figure 4a), using multiple prompts is expected to improve the precision of H-AIPW further. We confirm this observation in Appendix B.3, showing that H-AIPW precision improves with more prompts across several randomized studies.

## 6 Conclusion

We introduce H-AIPW, a novel estimator that can improve the efficiency of randomized experiments by integrating predictions from multiple foundation models. Our empirical results on social science data demonstrate that H-AIPW improves precision, especially in sample-constrained settings, without compromising validity of the inference. This approach holds significant promise in fields such as medicine, where leveraging well-curated foundation models could substantially lower the costs of clinical trials. However, a key limitation of H-AIPW is its reliance on the underlying foundation models: achieving meaningful gains in precision requires these models to be accurate and well-aligned with the experimental domain of interest. Further, finite-sample covariance estimation can be unstable when the number of models is large relative to the sample size, which may lead to undercoverage.

## Acknowledgments and Disclosure of Funding

PDB was supported by the Hasler Foundation grant number 21050 and the Ermenegildo Zegna Founder's Scholarship. JA was supported by the ETH AI Center. KD was supported by the ETH AI Center and the ETH Foundations of Data Science. This work was supported in part by National Library of Medicine (NLM) award R01LM013616; National Heart, Lung, and Blood Institute(NHLBI) award R01HL136708; and Patient-Centered Outcomes Research Institute (PCORI) award ME-2021C2-22365. The content is solely the responsibility of the authors and does not represent the official views of NLM, NHLBI, PCORI, PCORI's Board of Governors or PCORI's Methodology Committee.

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

# Appendices

The following appendices provide deferred proofs, ablation studies, and experimental details.

**Table of contents**

# A  Methodology

## A.1  Proofs

### A.1.1  Proof of Proposition 1

We adapt here a classic result from the semiparametric inference literature to our specific setting where the probability of treatment is known by design. For clarity, we refer to $\widehat{\theta}_{\text{AIPW}}$ as $\widehat{\theta}$.

Let us define the summand of the AIPW estimator for a fixed function $h$ as:

$$\psi_i(h) = \left( \frac{A_i}{\pi_1}(Y_i - h(X_i, 1)) + h(X_i, 1) \right) - \left( \frac{1 - A_i}{\pi_0}(Y_i - h(X_i, 0)) + h(X_i, 0) \right).$$

We can then decompose the estimation error of the AIPW estimator as follows:

$$\sqrt{n}(\widehat{\theta}(\widehat{h}) - \theta) = \underbrace{\sqrt{n}(\widehat{\theta}(h^\dagger) - \theta)}_{:=T_1} + \underbrace{\sqrt{n}(\widehat{\theta}(\widehat{h}) - \widehat{\theta}(h^\dagger))}_{:=T_2}.$$

The first term, $T_1$, is an average of i.i.d. random variables with mean zero and finite variance. Therefore, by the Central Limit Theorem, we have:

$$\sqrt{n}(\widehat{\theta}(h^\dagger) - \theta) = \sqrt{n} \left( \frac{1}{n} \sum_{i=1}^n \psi_i(h^\dagger) - \theta \right) \rightsquigarrow \mathcal{N}(0, V_{h^\dagger}),$$

where the asymptotic variance is given by $V_{h^\dagger} = \mathbb{E}[(\psi_i(h^\dagger) - \theta)^2]$.

**Bounding the remainder term**    We need to show that the second term $T_2$ is asymptotically negligible, that is $T_2 = o_{\mathbb{P}^*}(1)$.

We can rewrite this term as:

$$T_2 = \sqrt{n}(\widehat{\theta}(\widehat{h}) - \widehat{\theta}(h^\dagger)) = \frac{1}{\sqrt{n}} \sum_{i=1}^n \left( \psi_i(\widehat{h}) - \psi_i(h^\dagger) \right).$$

Further, with some simple algebra we can decompose the difference in the influence functions as:

$$\frac{1}{\sqrt{n}} \sum_{i=1}^n (\psi_i(\widehat{h}) - \psi_i(h^\dagger)) = \frac{1}{\sqrt{n}} \sum_{i=1}^n \left( \frac{A_i - \pi_1}{\pi_1} \right) (h^\dagger(X_i, 1) - \widehat{h}(X_i, 1))$$

$$- \frac{1}{\sqrt{n}} \sum_{i=1}^n \left( \frac{A_i - \pi_1}{1 - \pi_1} \right) (\widehat{h}(X_i, 0) - h^\dagger(X_i, 0))$$

Now, we will show that both terms in the sum above are asymptotically negligible. We focus our proof on the first term; the second follows from symmetric arguments.

Let $Z_i = (X_i, A_i, Y_i)$ and $\mathbb{P}_n$ denote the empirical measure over $Z_1, \ldots, Z_n$, and define the following functions:

$$f(Z_i) := \frac{A_i - \pi_1}{\pi_1} h^\dagger(X_i, 1) \ \text{ and } \ \widehat{f}(Z_i) := \frac{A_i - \pi_1}{\pi_1} \widehat{h}(X_i, 1).$$

We can rewrite the first term as:

$$\frac{1}{\sqrt{n}} \sum_{i=1}^n \left( \frac{A_i - \pi_1}{\pi_1} \right) (h^\dagger(X_i, 1) - \widehat{h}(X_i, 1)) = \sqrt{n} \, (\mathbb{P}_n - \mathbb{P})(f - \widehat{f}),$$

where we use the fact that $\mathbb{P}(f - \widehat{f}) = 0$, since the treatment probability is known. Given that $\widehat{h}$ is estimated from an independent sample, conditioning on $\widehat{h}$ the variables $\{f(Z_i) - \widehat{f}(Z_i)\}_{i=1}^n$ are i.i.d. with mean 0 and variance $\|\widehat{f} - f\|_{L^2(\mathbb{P})}^2$. Hence, by Chebyshev,

$$(\mathbb{P}_n - \mathbb{P})(\widehat{f} - f) = O_{\mathbb{P}^*} \left( \frac{\|\widehat{f} - f\|_{L_2(\mathbb{P})}}{\sqrt{n}} \right) = o_{\mathbb{P}^*} \left( \frac{1}{\sqrt{n}} \right),$$

where it follows from assumptions that $\|\widehat{f} - f\|_{L_2(\mathbb{P})} = o_{\mathbb{P}^*}(1)$. Therefore, we have that $T_2 = o_{\mathbb{P}^*}(1)$.

### A.1.2 Proof of Theorem 2

Recall that $\Sigma := \text{Cov}[(\psi(h^\dagger), \ldots, \psi(f_k))^\top]$ and define the oracle weights as $\lambda^\star = \underset{\lambda \in \Lambda}{\arg\min}\, \lambda^\top \Sigma \lambda$. The corresponding oracle estimator is then

$$\widehat{\theta}_{\lambda^\star} = \lambda_1^\star \widehat{\theta}_{\text{AIPW}}(\widehat{h}) + \sum_{j=1}^{k} \lambda_{j+1}^\star \widehat{\theta}_{\text{AIPW}}(f_j).$$

We now prove the theorem in the following three steps.

First, we observe that $\widehat{\theta}_{\lambda^\star}$ can also be written as

$$\widehat{\theta}_{\lambda^\star} = \widehat{\theta}_{\text{AIPW}}\left(\lambda_1^\star \widehat{h} + \sum_{j=1}^{k} \lambda_{j+1}^\star f_j\right),$$

since the constraint set is $\Lambda = \{\lambda \in \mathbb{R}^{k+1} : \sum_{j=1}^{k+1} \lambda_i = 1\}$. Further, it follows from assumptions that $\lambda_1^\star \widehat{h} + \sum_{j=1}^{k} \lambda_{j+1}^\star f_j$ is also an outcome function estimator that satisfies the conditions in Proposition 1, therefore $\widehat{\theta}_{\lambda^\star}$ is consistent and asymptotically normal, i.e. it holds that

$$\sqrt{n}(\widehat{\theta}_{\lambda^\star} - \theta) \rightsquigarrow \mathcal{N}(0, V_{\lambda^\star}), \quad \text{where} \quad V_{\lambda^\star} = \lambda^{\star\top} \Sigma \lambda^\star.$$

Second, we show that the asymptotic variance $V_{\lambda^\star}$ satisfies

$$V_{\lambda^\star} \leq \Sigma_{jj} \quad \text{for} \quad j = 1, \ldots, k+1.$$

By construction, the oracle weights $\lambda^\star$ minimize $\lambda^\top \Sigma \lambda$, ensuring $\widehat{\theta}_{\lambda^\star}$ attains the smallest asymptotic variance among all convex combinations of the initial estimators:

$$\left\{\widehat{\theta}_\lambda := \lambda_1 \widehat{\theta}_{\text{AIPW}}(\widehat{h}) + \sum_{j=1}^{k} \lambda_{j+1} \widehat{\theta}_{\text{AIPW}}(f_j) \,\Big|\, \lambda \in \Lambda\right\}.$$

Moreover, since $\lambda^\star$ is defined as the minimizer of $\lambda^\top \Sigma \lambda$ subject to $\mathbf{1}^\top \lambda = 1$, it holds that for any canonical vector $e_j \in \mathbb{R}^{k+1}$ (which corresponds to using the $j$th estimator alone) we have

$$V_{\lambda^\star} = \lambda^{\star\top} \Sigma \lambda^\star \leq e_j^\top \Sigma e_j = \Sigma_{jj}, \quad \text{for} \quad j = 1, \ldots, k+1.$$

Thus, the asymptotic variance of the hybrid estimator is no larger than that of any individual estimator.

Third, we observe that $\widehat{\theta}_{\widehat{\lambda}}$ and $\widehat{\theta}_{\lambda^\star}$ are asymptotically equivalent. Since $\|\widehat{\Sigma} - \Sigma\|_{\text{op}} \xrightarrow{p} 0$ and $\Sigma$ is nonsingular, the continuous mapping theorem implies

$$\widehat{\lambda} = \frac{\widehat{\Sigma}^{-1}\mathbf{1}}{\mathbf{1}^\top \widehat{\Sigma}^{-1}\mathbf{1}} \xrightarrow{p} \frac{\Sigma^{-1}\mathbf{1}}{\mathbf{1}^\top \Sigma^{-1}\mathbf{1}} = \lambda^\star.$$

Finally, using Slutsky's theorem, we get:

$$\sqrt{n}(\widehat{\theta}_{\widehat{\lambda}} - \theta) = \sqrt{n}(\widehat{\theta}_{\lambda^\star} - \theta) + o_{\mathbb{P}^*}(1) \rightsquigarrow \mathcal{N}(0, V_{\lambda^\star}),$$

which completes the proof.

## A.2 Connection with prediction-powered inference

To further study the connection and differences with prediction-powered inference (PPI) [1], it is instructive to consider the simpler problem of estimating the counterfactual mean, $\mathbb{E}[Y(1)]$[1]. For this case, a variant of PPI, referred to as PPI++ [2], can be shown to be equivalent to an AIPW estimator.

The standard difference in mean estimator is the sample mean of outcomes for the treated group:

$$\widehat{\theta}_{\text{DM}} = \frac{1}{n_1} \sum_{i:A_i=1} Y_i, \text{ where } n_a = \sum_{i=1}^{n} \mathbb{I}\{A_i = a\}.$$

PPI++ improves the difference in mean estimator by using predictions from a black-box model $f$:

$$\widehat{\theta}_{\text{PPI++}} = \frac{1}{n_1} \sum_{i:A_i=1} Y_i + \lambda \left( -\frac{1}{n_1} \sum_{i:A_i=1} f(X_i) + \frac{1}{n_0} \sum_{i:A_i=0} f(X_i) \right),$$

where the power-tuning parameter $\lambda$ is chosen to minimize the variance. Crucially, for $\lambda = \frac{n_0}{n_1+n_0}$, assuming exact randomization, i.e. $\pi_1 = n_1/n$, we have equivalence with the AIPW estimator for the counterfactual mean,

$$\widehat{\theta}_{\text{PPI++}} = \frac{1}{n} \sum_{i=1}^{n} \left( \frac{A_i(Y_i - f(X_i))}{\pi_1} + f(X_i) \right) = \widehat{\theta}_{\text{AIPW}}(f).$$

A few remarks are in order.

- PPI++ replaces the estimated outcome regression with a black-box model $f$. However, when $f$ is not equivalent to the outcome regression $\mathbb{E}[Y \mid X, A = 1]$, the resulting estimator will not be efficient. In other words, $\widehat{\theta}_{\text{PPI++}}$ will not achieve the smallest asymptotic variance among the regular estimators of the counterfactual mean. Concurrent work by Ji et al. [30] similarly identifies this limitation and proposes a recalibrated version of PPI to overcome it. By contrast, the AIPW estimator will achieve the smallest possible asymptotic variance, assuming that the outcome regression estimator is consistent in $L_2$-norm. This condition is easier to satisfy in the setting of randomized experiments, since we can use flexible machine-learning models and still have valid inference as a consequence of Proposition 1. In particular, our H-AIPW estimator is guaranteed to have asymptotic variance no greater than the standard AIPW estimator (Theorem 2), and thus can be efficient even if the black-box model $f$ is arbitrarily biased.

- Extending PPI and PPI++ to average treatment effect estimation is not straightforward. To do so, Poulet et al. [40] proposes the following estimator:

$$\widehat{\theta}_{\text{PPCT}} := \frac{1}{n_1} \sum_{A_i=1} (Y_i - \lambda f(X_i)) - \frac{1}{n_0} \sum_{A_i=0} (Y_i - \lambda f(X_i)).$$

However, a key limitation of the above estimator is that it forces both outcome regressions, that is $\mathbb{E}[Y \mid X = x, A = 1]$ and $\mathbb{E}[Y \mid X = x, A = 0]$, to be replaced with the same black-box model $f$. This is particularly problematic when the treatment has a significant effect on the outcome, as a single model $f$ will fail to accurately capture both outcome regressions. In contrast, our approach allows for different black-box models $f_1$ and $f_0$ to be plugged-in for the treated and control group, respectively.

- PPI and its variants cannot integrate multiple competing foundation models. This is a key limitation in the causal inference setting, as model selection is a non-trivial task due to the missingness of potential outcomes. Moreover, it is unclear whether they can be extended to do so, as constructing a consistent estimate of the covariance matrix $\Sigma$ poses a major hurdle. In contrast, our approach offers a simple way to estimate the covariance matrix $\Sigma$ by exploiting the linear structure of the AIPW estimators.

---

[1]We refer the reader to Xu et al. [51] for a discussion of the connections between AIPW and PPI.

## A.3 Dependency of the variance term on the estimation error $\|\widehat{h} - h^\star\|_{L_2(\mathbb{P})}$

As mentioned in Section 3.1, in small sample regimes the variance of the AIPW estimator crucially depends on the estimation error $\|\widehat{h} - h^\star\|_{L_2(\mathbb{P})}$. For completeness, we formalize here this dependency by bounding the excess variance of the AIPW estimator that arises from using $\widehat{h}$ instead of $h^\star$.

**Lemma 1.** *For any outcome regression $\widehat{h}$ estimated from an independent sample, we have*

$$\mathrm{Var}(\sqrt{n}\,\widehat{\theta}_{\mathrm{AIPW}}(\widehat{h})) - \mathrm{Var}(\sqrt{n}\,\widehat{\theta}_{\mathrm{AIPW}}(h^\star)) = \mathbb{E}\left[\left(\sqrt{\frac{\pi_1}{\pi_0}}\left(\widehat{h}(X,0) - h^\star(X,0)\right) + \sqrt{\frac{\pi_0}{\pi_1}}\left(\widehat{h}(X,1) - h^\star(X,1)\right)\right)^2\right].$$

*And thus, it holds that*

$$\mathrm{Var}(\sqrt{n}\,\widehat{\theta}_{\mathrm{AIPW}}(\widehat{h})) - \mathrm{Var}(\sqrt{n}\,\widehat{\theta}_{\mathrm{AIPW}}(h^\star)) \leq \frac{1}{\pi_0}\|\widehat{h}(.,0) - h^\star(.,0)\|_{L_2(\mathbb{P})}^2 + \frac{1}{\pi_1}\|\widehat{h}(.,1) - h^\star(.,1)\|_{L_2(\mathbb{P})}^2.$$

*Proof of Lemma 1.* Note that by the unbiasedness of the AIPW estimator, as well as the independence of the samples used to compute the outcome regression $\widehat{h}$, the excess variance equals:

$$n\,\mathrm{Var}(\widehat{\theta}_{\mathrm{AIPW}}(\widehat{h})) - n\,\mathrm{Var}(\widehat{\theta}_{\mathrm{AIPW}}(h^\star)) = \mathbb{E}\left[(\psi(\widehat{h}) - \theta)^2 - (\psi(h^\star) - \theta)^2\right]$$

$$= \mathbb{E}\left[\underbrace{2\triangle\psi\,(\psi(h^\star) - \theta)}_{=:T_1} + \underbrace{\triangle\psi^2}_{=:T_2}\right],$$

with $\triangle\psi = \psi(\widehat{h}) - \psi(h^\star)$. We bound the two terms $T_1$ and $T_2$ separately. Recall that by definition,

$$\psi(h) := \left(\frac{A}{\pi_1}(Y - h(X,1)) + h(X,1)\right) - \left(\frac{1-A}{\pi_0}(Y - h(X,0)) + h(X,0)\right),$$

and thus,

$$\triangle\psi = \triangle h_1\left(1 - \frac{A}{\pi_1}\right) - \triangle h_0\left(1 - \frac{1-A}{\pi_0}\right), \quad \text{with } \triangle h_i(X) := \widehat{h}(X,i) - h^\star(X,i).$$

which does not depend on $Y$. Hence, taking the expectation first over $Y$ for $T_1$ yields:

$$T_1 = 2\,\mathbb{E}\left[\triangle\psi\,(h^\star(X,1) - h^\star(X,0) - \theta)\right],$$

where we used the fact that $\mathbb{E}\left[Y|X,A\right] = h^\star(X,A)$. Finally, since $\mathbb{E}\left[\triangle\psi|X\right] = 0$, we also obtain that $T_1 = 0$.

Next, to bound the second term $T_2$, we can write:

$$T_2 = \mathbb{E}\left[\triangle h_1^2\left(1 - \frac{A}{\pi_1}\right)^2 + \triangle h_0^2\left(1 - \frac{1-A}{\pi_0}\right)^2 - 2\triangle h_1\triangle h_0\left(1 - \frac{1-A}{\pi_0}\right)\left(1 - \frac{A}{\pi_1}\right)\right].$$

A straightforward computation (using $\pi_1 = 1 - \pi_0$) yields:

$$\mathbb{E}\left[\triangle h_1^2\left(1 - \frac{A}{\pi_1}\right)^2\right] = \mathbb{E}\left[\triangle h_1^2\right]\left(\frac{1}{1 - \pi_0} - 1\right)$$

$$\text{and} \quad \mathbb{E}\left[\triangle h_0^2\left(1 - \frac{1-A}{\pi_0}\right)^2\right] = \mathbb{E}\left[\triangle h_0^2\right]\left(\frac{1}{\pi_0} - 1\right)$$

$$\text{and} \quad -\mathbb{E}\left[2\triangle h_1\triangle h_0\left(1 - \frac{1-A}{\pi_0}\right)\left(1 - \frac{A}{\pi_1}\right)\right] = 2\mathbb{E}\left[\triangle h_0\triangle h_1\right].$$

As a result, we obtain:

$$T_2 = \mathbb{E}\left[\left(\sqrt{\frac{1 - \pi_0}{\pi_0}}\triangle h_0 + \sqrt{\frac{\pi_0}{1 - \pi_0}}\triangle h_1\right)^2\right],$$

which completes the proof. $\qquad\square$

Table 2: Coverage probability comparison of H-AIPW against baseline estimators (PPCT, DM, AIPW, PROCOVA) across several randomized experiments. We report the empirical coverage of each estimator's $1 - \alpha$ confidence interval. The coverage probability is averaged over $R = 10000$ subsampling repetitions at sample sizes $n = 100$ and $n = 200$. The nominal level is set at $\alpha = 0.05$. We implement H-AIPW by integrating predictions from three LLMs: GPT-4o, Claude 3.5 Haiku, and LLaMA 3 70B.

| | Melin et al. (2022) | | Silverman et al. (2022) | | Kennedy et al. (2020) | | Fahey et al. (2023) | |
| --- | --- | --- | --- | --- | --- | --- | --- | --- |
| **Estimator** | $n = 100$ | $n = 200$ | $n = 100$ | $n = 200$ | $n = 100$ | $n = 200$ | $n = 100$ | $n = 200$ |
| H-AIPW | 96.2 | 98.4 | 96.4 | 98.4 | 94.4 | 95.9 | 94.5 | 95.6 |
| PPCT | 96.4 | 98.4 | 96.7 | 98.7 | 95.0 | 96.5 | 94.9 | 95.7 |
| PROCOVA | 96.4 | 98.4 | 96.4 | 97.7 | 94.9 | 96.2 | 95.2 | 95.8 |
| AIPW (standard) | 96.5 | 98.5 | 96.3 | 97.7 | 94.8 | 96.1 | 95.3 | 96.0 |
| AIPW (boosting) | 95.6 | 97.2 | 95.6 | 96.6 | 94.4 | 95.0 | 94.1 | 94.6 |
| DM | 96.6 | 98.5 | 96.9 | 98.7 | 95.4 | 96.5 | 95.8 | 96.7 |
| | Caprariello et al. (2013) | | Brandt (2013) | | Haaland et al. (2023) | | Shuman et al. (2024) | |
| **Estimator** | $n = 100$ | $n = 200$ | $n = 100$ | $n = 200$ | $n = 100$ | $n = 200$ | $n = 100$ | $n = 200$ |
| H-AIPW | 96.8 | 99.3 | 95.7 | 97.2 | 95.0 | 96.2 | 95.3 | 96.2 |
| PPCT | 97.2 | 99.3 | 96.1 | 97.7 | 95.0 | 96.3 | 94.9 | 96.4 |
| PROCOVA | 97.5 | 99.4 | 96.3 | 97.6 | 95.2 | 96.2 | 95.0 | 95.8 |
| AIPW (standard) | 97.6 | 99.4 | 96.3 | 97.4 | 95.2 | 96.4 | 95.1 | 96.1 |
| AIPW (boosting) | 96.7 | 98.6 | 95.5 | 95.0 | 94.3 | 95.4 | 93.4 | 95.1 |
| DM | 97.4 | 99.4 | 96.6 | 98.1 | 95.3 | 96.3 | 95.5 | 96.5 |

# B  Additional experiments

We present here additional ablations of our method. The results reinforce the general trends observed in the main experiments: H-AIPW achieves better precision than the baselines while maintaining comparable coverage. Ablation studies provide insight into the number of models that can be incorporated into our estimator without significantly compromising validity (due to finite sample effects), and they offer further evidence of the advantages of increasing inference-time compute.

## B.1  Empirical evaluation of coverage probability

To benchmark validity, for each estimator, we compute the fraction of confidence intervals containing the average treatment effect:

$$\text{Coverage} = \frac{1}{R} \sum_{r=1}^{R} \mathbb{I}\{\theta \in \mathcal{C}_r^\alpha\},$$

where $\mathcal{C}_r^\alpha$ is the confidence interval obtained from the dataset $\mathcal{D}_r$ and $\theta$ is the difference in means ATE estimate from the full study dataset. While this is not necessarily the true ATE, it serves as the best available proxy in the context of real randomized experiments. Table 2 shows that H-AIPW consistently achieves coverage probability close to the nominal 95% level across all studies and both sample sizes. Importantly, this indicates that the variance reductions observed in Table 1 do not come at the expense of statistical validity.

## B.2  Impact of adding more foundation models on statistical precision

In this section, we study the impact of increasing the number of models in H-AIPW. Specifically, Algorithm 1 requires integrating predictions from multiple foundation models, which are combined with the standard AIPW to minimize the variance of the resulting estimator. In Figure 5, we show how increasing the number of language models from 1 to 7 affects the precision and validity of H-AIPW in the study by Fahey et al. [22]. Models are incorporated in the estimator sequentially, starting from those with the lowest mean squared error (MSE) (i.e. LLaMA 3 70B) to those with the

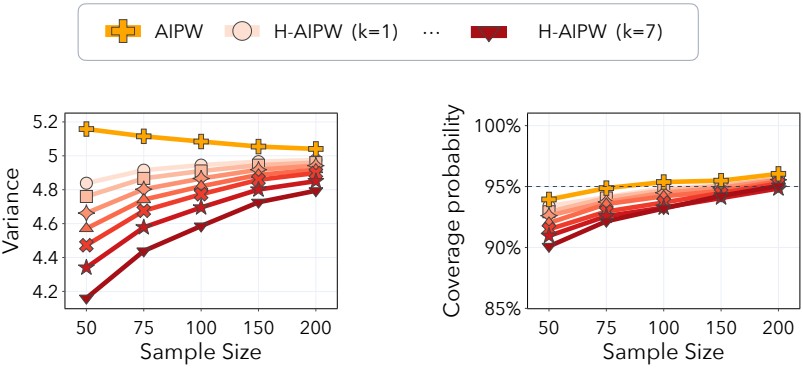

Figure 5: Impact of increasing the number of models in H-AIPW on precision and validity in the study by Fahey et al. [22]. Models are sequentially incorporated based on their mean squared error (MSE), starting with LLaMA 3 70B (lightest red, $k = 1$) and ending with Gemma 2 27B (darkest red, $k = 7$), following Figure 4a. The left panel shows the empirical variance, while the right panel shows empirical coverage. The standard AIPW estimator is included for reference. Each experiment is averaged over $R = 10k$ repetitions, with significance level set to $\alpha = 0.05$.

highest (stopping at Gemma 2 27B), following Figure 4a. We also include the standard AIPW (linear) estimator for reference.

Increasing the number of models improves precision compared to the standard AIPW estimator. In the setting with 50 samples, a single model improves variance by approximately $6\%$, while using 4 models increases this gain to nearly $12\%$, and 7 models yield an improvement of around $16\%$. However, the marginal benefits diminish with larger sample sizes: at 200 observations, the variance difference between using 1 and 7 models shrinks to $4\%$. However, adding more models weakens empirical coverage. With 50 samples, combinations of 5 to 7 models exhibit undercoverage of $2\%$–$4\%$ relative to AIPW, failing to reach the nominal $95\%$ coverage until the sample size reaches 200. In contrast, combinations of 1 to 3 models maintain coverage levels comparable to AIPW.

Intuitively, the undercoverage observed in Figure 5 is driven by finite-sample error in estimating the covariance matrix. As the number of models increases, the covariance matrix is harder to estimate reliably from limited data. This, in turn, leads to a systematic underestimation of the combined estimator's variance and hence to undercoverage. Such effects have been formally proven in recent "no free lunch" results on prediction-powered inference [38]. A simple remedy is to use sample-splitting or cross-fitting when estimating the covariance matrix, ensuring that the same data are not reused for both covariance matrix estimation and inference.

Practitioners should therefore carefully determine both the number of models to include in the ensemble and whether to adopt sample-splitting or cross-fitting. In our experiments, with moderately large samples ($n = 100$ and $n = 200$) and only three outcome models, we did not observe undercoverage, suggesting that the additional estimation error is negligible and standard plug-in inference is sufficient. By contrast, with smaller samples or ensembles of several models, the covariance matrix's estimation error becomes non-negligible, and cross-fitting is recommended to ensure valid coverage.

## B.3   Impact of inference-time compute on statistical precision

In Section 5.2, we showed that increasing inference-time compute improves the precision of H-AIPW: more prompts generally reduce mean squared error (MSE) of the foundation model predictions, which in turn lowers the estimator's variance. For completeness, Figure 6 visualizes the relationship between the number of prompts, MSE, and variance.

We present results for three studies—Brandt [7], Silverman et al. [46], Kennedy and Horne [33]—using H-AIPW with predictions from GPT-4o. Figures 6a to 6c show the empirical estimate of the variance as a function of the number of prompts, while Figures 6d to 6f illustrate the corresponding

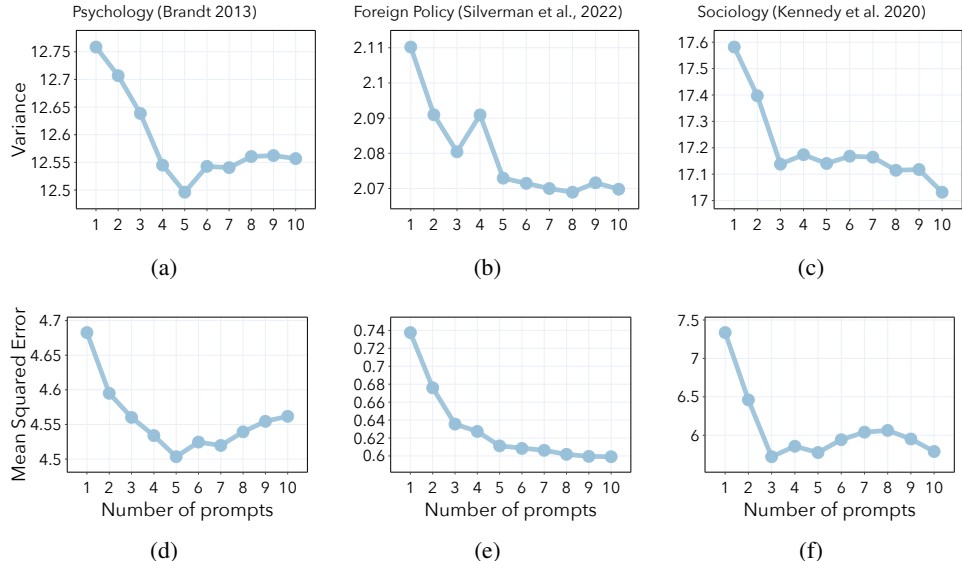

Figure 6: Impact of the number of prompts on the empirical variance and MSE. Results are reported for studies by Brandt [7], Silverman et al. [46], Kennedy and Horne [33]. We randomly subsample each study to obtain a sample size $n = 50$ and report the average over $R = 10k$ repetitions for each metric. **(First row)** Reduction in variance as the number of prompts increases. **(Second row)** Reduction in MSE as the number of prompts increase. These results suggest that increasing inference-time compute improves the precision of H-AIPW by reducing the MSE of the foundation model predictions.

changes in MSE. The findings reinforce the conclusions from the main text: increasing inference-time compute through multiple prompts generally reduces the variance of H-AIPW.

# C   Experimental details

## C.1   Implementation details

For all experiments, we first select the five features most correlated with the outcome variable. The AIPW estimator implements cross-fitting with 30 folds, using ridge regression with regularization $\lambda = 1.0$ in the standard case and XGBoost with default hyperparameters in the boosting case. For PPCT, we follow the implementation by Poulet et al. [40], using GPT-4o's predictions for the control scenario as the prognostic score. We implement PROCOVA using an AIPW estimator whose outcome regression estimator is augmented with a smart covariate, i.e. the prediction of GPT-4o for both arms. The coefficients for the optimal combination are computed using standard `Python` libraries. Finally, the DM estimator requires no hyperparameter tuning. We compute the ground-truth ATE for all studies using the DM estimator on the full study with sample size $N$.

**Implementation of H-AIPW**   Our estimator integrates synthetic outcomes generated by multiple LLMs. Unless stated otherwise, we use predictions from LLaMA 3 70B, GPT-4o, and Claude 3.5 Haiku for all experiments in Section 5.1. Additional models, such as Gemma 2, Grok 2, and Gemini 1.5 Flash, are used in specific cases, e.g. Section 5.2. We leverage both proprietary and open-source LLMs. For open-source models, we apply nucleus sampling with a temperature of 1.2, top-p of 0.9, and a maximum of 100 new tokens. For proprietary models, we use default decoding settings, except for Claude 3.5 Haiku, where we set the temperature to 1. In summary, H-AIPW extends the classic AIPW estimator by incorporating multiple AIPW estimators that integrate LLM predictions; see Algorithm 1 for full details.

**Reproducing Figure 2**   We randomly subsample each study with a sample size of $n = 75$ and compute the average confidence interval over 1k repetitions using the *standard* AIPW estimator with linear regression. Then, we obtain $n_{\text{H-AIPW}}$ by progressively reducing $n$ until the average confidence interval from H-AIPW (using GPT-4o, LLaMa 3 70B, and Claude 3.5 Haiku) matches or exceeds the standard AIPW confidence interval. The percentage reduction in sample size is then computed as:

$$100 \left( 1 - \frac{n_{\text{H-AIPW}}}{n} \right).$$

## C.2   Preprocessing of scientific studies and prompt design

In this section, we describe the preprocessing steps, selected outcomes, and control and treatment scenarios for the studies used in our experiments. We also provide an example prompt, including both system and user components, used to query the LLMs. The studies are sourced from the Time-sharing Experiments for the Social Sciences (TESS) repository, with findings published in peer-reviewed journals. These studies span various fields, demonstrating the versatility of our methodology.

### C.2.1   Can Factual Misperceptions be Corrected? An Experiment on American Public Fears of Terrorism [46]

**Abstract:** An American's yearly chance of being killed by a terrorist attack sits at about 1 in 3.5 million. Yet over 40% of the American public consistently believes that they or their family members are likely to be the victim of a terror attack. Can these inflated estimates of the risks of terrorism be brought closer to reality? With trillions of dollars spent on the War on Terror since 9/11, this question is not just theoretically but practically important. In order to investigate, we field a nationally representative survey experiment containing a brief vignette with corrective information about the actual risks of terrorism vs. other dangers facing Americans. Additionally, we vary whether there is a political elite endorsement accompanying the information, with either a Democratic politician, Republican politician, or senior military officer driving home the message.

**Data availability:** The study is publicly available at: `https://tessexperiments.org/study/silverman1035`

**Data pre-processing:** The primary outcome variable is `Q5`. The treatment condition is defined as `P_TESS031 = 1` (corrective information), and the control condition is defined as `P_TESS031 = 0` (no corrective information). The following variables are included as covariates: `PARTYID7`, `IDEO`, `RELIG`, `ATTEND`, `GENDER`, `AGE`, `RACETHNICITY`, `EDUC4`, `INCOME`. The final processed dataset contains $n = 503$ observations.

**Prompting details:** An example prompt is provided below.

---

**Example Prompt**

**System Prompt:**

> You are a 33-year-old, ethnicity White, gender Male, strong Democrat. You hold very liberal views and college education. Additionally, your religion is Catholic, and you attend religious services nearly every week. Your household has a yearly income of $75,000 to $84,999. Your answer must be a single integer without additional text, in JSON format with a key-value pair.

**Treatment Condition:**

> The number of people who say that acts of terrorism against Americans are imminent is up 3% from last year, according to a new poll released this week. In the wake of attacks in San Bernardino, Orlando, Paris, and London, the Pew Research Center found that 63% of Americans think major terrorist attacks are likely to occur soon on American soil. Government officials have echoed these concerns. "We are issuing a new advisory that the terror threat is now elevated across the country," said Undersecretary for Homeland Security Stephen Krause. "We have to remain vigilant and we have to stay alert. Terrorists can strike anytime, anywhere."
>
> But does terrorism really pose a critical threat to us? Below is a figure showing the average American's risk of death from different sources. As can be seen, around 90 Americans are killed each year by terrorism on U.S. soil. This means the risk of being a victim of terrorism in a given year is about 1 in 3.5 million. In comparison, the risk of being killed by cancer is 1 in 540, the risk of being killed in a car accident is 1 in 8,000, and the chance of being killed by your own home appliances is 1 in 1.5 million. These numbers provide some essential context when thinking about the different threats to our public safety.

**Control Condition:**

> The number of people who say that acts of terrorism against Americans are imminent is up 3% from last year, according to a new poll released this week. In the wake of attacks in San Bernardino, Orlando, Paris, and London, the Pew Research Center found that 63% of Americans think major terrorist attacks are likely to occur soon on American soil. Government officials have echoed these concerns. "We are issuing a new advisory that the terror threat is now elevated across the country," said Undersecretary for Homeland Security Stephen Krause. "We have to remain vigilant and we have to stay alert. Terrorists can strike anytime, anywhere."

**Question:**

> How likely do you think it is that another terrorist attack causing large numbers of American lives to be lost will happen in the near future? Choose an integer between 1 (very likely) and 5 (not likely at all).

---

### C.2.2 Cancel Culture for Friends, Consequence Culture for Enemies: The Effects of Ideological Congruence on Perceptions of Free Speech [22]

**Abstract:** Political scientists have long been interested in the effects that media framings have on support or tolerance for controversial speech. In recent years, the concept of cancel culture has

complicated our understanding of free speech. In particular, the modern Republican Party under Donald Trump has made "fighting cancel culture" a cornerstone of its electoral strategy. We expect that when extremist groups invoke cancel culture as a reason for their alleged censorship, support for their free speech rights among Republicans should increase. We use a nationally representative survey experiment to assess whether individuals' opposition to cancel culture is principled or contingent on the ideological identity of the speaker. We show that framing free speech restrictions as the consequence of cancel culture does not increase support for free speech among Republicans. Further, when left-wing groups utilize the cancel culture framing, Republicans become even less supportive of those groups' free speech rights.

**Data availability:** The study is publicly available at: https://www.tessexperiments.org/study/faheyS78

**Data pre-processing:** The primary outcome variable is `CC_1`. The treatment condition is defined as `P_GROUP = 2` (safety reasons + cancel culture), and the control condition as `P_GROUP = 1` (safety reasons). The following variables are included as covariates: `PARTYID7`, `IDEO`, `RELIG`, `ATTEND`, `GENDER`, `AGE`, `HOME_TYPE`, `INCOME`. The final processed dataset contains $n = 998$ observations.

**Prompting details:** An example prompt is provided below.

---

### Example Prompt

**System Prompt:**

> You are a 35-year-old male, politically Democrat, holding liberal views. Additionally, your religion is Christianity, and you once or twice a month attend religious services. You reside in a building with two or more apartments, and your household has a yearly income of $85,000 to $99,999. You are responding to a scenario reflecting a debate involving college campus events and broader social issues.

**Treatment Condition:**

> We are now going to ask you to imagine you have read about the following scenario, describing a debate on a recent College Campus.
> **Local Group Denied Permit to Protest on Campus, Provoking Debate About "Cancel Culture"**
> A debate on the merits of free speech erupted recently when the student chapter of the controversial far-left group Antifa attempted to obtain a permit to conduct a demonstration on the main quad of Rutgers University in New Jersey. Citing safety concerns, the president of the organization in charge of Registered Student Organizations (RSOs) initially denied the organization the right to conduct their rally, arguing that their presence would endanger college students. They cited a recent incident in Berkeley, CA where three Antifa members and two bystanders were injured by rocks thrown in an altercation between the group and counter protesters. A member of the local Antifa group, Luke Vargas, is appealing the decision, arguing that the permit denial represented "cancel culture run amok," and the University was simply "afraid to hear the truth." When asked to comment, the University Ombudsman's Office promised that a final decision on whether the rally would be permitted would be made by this Thursday, three days before the march is scheduled to take place on Sunday.

**Control Condition:**

> We are now going to ask you to imagine you have read about the following scenario, describing a debate on a recent College Campus.
> **Local Group Denied Permit to Protest on Campus**
> A debate on the merits of free speech erupted recently when the student chapter of the controversial far-left group Antifa attempted to obtain a permit to conduct a demonstration on the main quad of Rutgers University in New Jersey. Citing safety concerns, the president of the organization in charge of

---

Registered Student Organizations (RSOs) initially denied the organization the right to conduct their rally, arguing that their presence would endanger college students. They cited a recent incident in Berkeley, CA where three Antifa members and two bystanders were injured by rocks thrown in an altercation between the group and counter protesters. A member of the local Antifa group, Luke Vargas, promised to bring an appeal to the desk of the University President. When asked to comment, the University Ombudsman's Office promised that a final decision on whether the rally would be permitted would be made by this Thursday, three days before the march is scheduled to take place on Sunday.

**Question:**

Generally speaking, do you agree or disagree with the following statement: "Cancel culture is a big problem in today's society." Reply using numbers between 1 (definitely agree) and 5 (definitely disagree).

### C.2.3 Beliefs about Racial Discrimination [28]

**Abstract:** This paper provides representative evidence on beliefs about racial discrimination and examines whether information causally affects support for pro-black policies. Eliciting quantitative beliefs about the extent of hiring discrimination against blacks, we uncover large disagreement about the extent of racial discrimination with particularly pronounced partisan differences. An information treatment leads to a convergence in beliefs about racial discrimination but does not lead to a similar convergence in support of pro-black policies. The results demonstrate that while providing information can substantially reduce disagreement about the extent of racial discrimination, it is not sufficient to reduce disagreement about pro-black policies.

**Data availability:** The study is publicly available at: https://www.tessexperiments.org/study/Haaland874

**Data pre-processing:** The primary outcome variable is Q2. The treatment condition is defined as GROUP = 1 (statistics of white-sounding and black-sounding names), and the control condition is defined as GROUP = 2 (statistics of white-sounding names). The following variables are included as covariates: PartyID7, INCOME, ATTEND, RELIG, GENDER, AGE, REGION9, RACETHNICITY. The final processed dataset contains $n = 1539$ observations.

**Prompting details:** An example prompt is provided below.

---

**Example Prompt**

**System Prompt:**

You are a 60-year-old, politically Independent, gender Female, ethnicity Hispanic. Additionally, your religion is just Christian and you never attend religious services. You live in a state of the West South Central region. Your household has a yearly income of $30,000 to $34,999. You are responding to a survey experiment collecting data on people's beliefs about racial discrimination and whether these beliefs affect people's views on affirmative action policies.

**Treatment condition:**

Researchers from Harvard University conducted an experiment to study racial discrimination in the labor market. They did so by sending out fictitious resumes to help-wanted ads in Boston newspapers. The resumes were exactly the same except for one thing: the name of the job applicant. Half of the resumes had typically white-sounding names like "Carrie" and "Todd". The other half of the resumes had typically black-sounding names like "Tanisha" and "Kareem". The idea was to make sure that the applicants were seen

---

as having identical qualifications, but that the employers would use the applicants' names to infer whether they were white or black. Resumes with white-sounding names had to be sent out on average 10 times to get one callback for an interview.

Further, the researchers found that resumes with black-sounding names on average had to be sent out 15 times to get one callback for an interview. Since resumes with white-sounding names on average only had to be sent out 10 times to get one callback for an interview, this means that employers were 50 percent more likely to give callbacks to applicants with white-sounding names compared to applicants with black-sounding names.

**Control condition:**

Researchers from Harvard University conducted an experiment to study racial discrimination in the labor market. They did so by sending out fictitious resumes to help-wanted ads in Boston newspapers. The resumes were exactly the same except for one thing: the name of the job applicant. Half of the resumes had typically white-sounding names like "Carrie" and "Todd". The other half of the resumes had typically black-sounding names like "Tanisha" and "Kareem". The idea was to make sure that the applicants were seen as having identical qualifications, but that the employers would use the applicants' names to infer whether they were white or black. Resumes with white-sounding names had to be sent out on average 10 times to get one callback for an interview.

**Question:**

In the United States today, do you think that racial discrimination against blacks in the labor market is a serious problem? Reply with a JSON numerical answer using one of these numbers: 1 (A very serious problem), 2 (A serious problem), 3 (A problem), 4 (A small problem), or 5 (Not a problem at all).

### C.2.4 Accidental Environmentalists: Examining the Effect of Income on Positive Social Evaluations of Environmentally-Friendly Lifestyles [33]

**Abstract:** Many US households have adopted behaviors aimed at reducing their environmental impact. Existing scholarship examines antecedent variables predicting engagement in these pro-environmental behaviors. But little research examines the effect of making efforts to reduce environmental impact on positive evaluations. Based on our qualitative pilot data, we suspect that income may be an important factor in the extent to which green lifestyles earn social approval. We predict that a household that reduces its environmental impact will be viewed more positively if that household has a high (rather than low) income. We manipulate household income (high vs low) and proenvironmental behavior (green vs typical). We then measure participants' approval of the household, how socially close they feel to the household, as well as their evaluations of the household's competence, morality, and environmental commitment. This research allows us to identify the bases for social approval of green lifestyles and examine how social approval for a household's green lifestyle varies with that household's income.

**Data availability:** The study is publicly available at: https://tessexperiments.org/study/kennedy1017

**Data pre-processing:** The primary outcome variable is Q5. The treatment condition is defined as P_TESS23 = 4 (green lifestyle), and the control condition is defined as P_TESS23 = 2 (typical lifestyle). The following variables are included as covariates: PartyID7, IDEO, ATTEND, GENDER, AGE. The final processed dataset contains $n = 1276$ observations.

**Prompting details:** An example prompt is provided below.

> **Example Prompt**
>
> **System Prompt:**
>
> You are a 45-year-old, lean Democrat, gender Female, and hold slightly conservative views. Additionally, you attend religious services several times a year. We are going to give you some information about a family. Please read the information very carefully, as we will be asking you questions about it. Your answer must be in JSON format with a single key-value pair.
>
> **Treatment condition:**
>
> A family with two children lives in a neighborhood nearby to yours. You chat with them sometimes when you see them in the neighborhood. As far as you can tell, they make a huge amount of money and seem to have plenty of extra money to spend. Their house is small and they often take public transit or walk to avoid driving. They also dry their clothes on a clothesline and don't have air conditioning in their home. This family has a much lower environmental impact than other people in their neighborhood.
>
> **Control condition:**
>
> A family with two children lives in a neighborhood nearby to yours. You chat with them sometimes when you see them in the neighborhood. As far as you can tell, they make very little money and seem to have no extra money to spend. Their house is small and they often take public transit or walk to avoid driving. They also dry their clothes on a clothesline and don't have air conditioning in their home. This family has a much lower environmental impact than other people in their neighborhood.
>
> **Question:**
>
> How much is the environment a high priority for this family? Choose an integer between 1 (not at all) and 11 (very much).

### C.2.5 To Do, to Have, or to Share? Valuing Experiences and Material Possessions by Involving Others [10]

**Abstract:** Recent evidence indicates that spending discretionary money with the intention of acquiring life experiences-events that one lives through-makes people happier than spending money with the intention of acquiring material possessions-tangible objects that one obtains and possesses. We propose and show that experiences are more likely to be shared with others, whereas material possessions are more prone to solitary use and that this distinction may account for their differential effects on happiness. In 4 studies, we present evidence demonstrating that the inclusion of others is a key dimension of how people derive happiness from discretionary spending. These studies showed that when the social-solitary and experiential-material dimensions were considered simultaneously, social discretionary spending was favored over solitary discretionary spending, whereas experiences showed no happiness-producing advantage relative to possessions. Furthermore, whereas spending money on socially shared experiences was valued more than spending money on either experiences enacted alone or material possessions, solitary experiences were no more valued than material possessions. Together, these results extend and clarify the basic findings of prior research and add to growing evidence that the social context of experiences is critical for their effects on happiness.

**Data availability:** The study is publicly available at: `https://www.tessexperiments.org/study/caprariello130`

**Data pre-processing:** The primary outcome variable is `Q7A`. The treatment condition is defined as `XTESS086 = 1` (spend money with people), and the control condition is defined as `XTESS086 = 2` (spend money alone). The following variables are included as covariates: `XPARTY7`, `XREL1`, `XREL2`, `XIDEO`, `PPAGE`, `PPGENDER`. The final processed dataset contains $n = 397$ observations.

**Prompting details:** An example prompt is provided below.

> **Example Prompt**
>
> **System Prompt:**
>
> > You are a 53-year-old, not so strong Republican, gender Male, and hold moderate views. Additionally, regarding religion you are Buddhist and you more than once a week attend religious services. You are responding to a survey on how you spend your discretionary money. Your answer must be a single integer without additional text, in JSON format with a key-value pair.
>
> **Treatment condition:**
>
> > We are interested in ways you spend your discretionary money. Discretionary money refers to money that is spent on anything that is NOT essential to basic activity (that is, essentials refer to things like tuition and textbooks, groceries, transportation, rent, gas for a car, health care, etc.). We'd like you to answer the questions that follow for money that you spent on something discretionary. Please think of the last time you spent at least $10 (but no more than $10,000) of your discretionary money in order TO DO SOMETHING WITH AT LEAST ONE OTHER PERSON. The primary focus of this expense should have been on an activity – doing something with at least one other person – and not on buying something that could be kept. Maybe you bought tickets to see a movie with some people, maybe you paid to visit an art museum with friends, maybe you and some other people went to a spa together . . . any of these would be legitimate examples of spending money to do something with others.
>
> **Control condition:**
>
> > We are interested in ways you spend your discretionary money. Discretionary money refers to money that is spent on anything that is NOT essential to basic activity (that is, essentials refer to things like tuition and textbooks, groceries, transportation, rent, gas for a car, health care, etc.). We'd like you to answer the questions that follow for money that you spent on something discretionary. Please think of the last time you spent at least $10 (but no more than $10,000) of your discretionary money in order TO DO SOMETHING BY YOURSELF. The primary focus of this expense should have been on an activity – doing something by yourself – and not on buying something that could be kept. Maybe you bought a ticket to see a movie by yourself, maybe you paid to enter an art museum, maybe you went to a spa by yourself . . . any of these would be legitimate examples of spending money to do something by yourself.
>
> **Question:**
>
> > Think about the last time you used your possession. To what extent did it help you feel loved and cared about? Reply with a JSON numerical answer using one of these numbers: 1 (not at all), 2 (slightly), 3 (moderately), 4 (very), or 5 (extremely).

### C.2.6 Onset and Offset Controllability in Perceptions and Reactions to Home Mortgage Foreclosures [7]

**Abstract:** The circumstances and rhetoric surrounding home foreclosures provide an ideal and timely backdrop for an extension of research on attributional judgments. While people face foreclosure for many reasons, the current debate surrounding the mortgage crisis has highlighted reasons that are either onset or offset controllable; that is, the initial cause, or the subsequent solution may be seen as controllable.In the current study, I examine how people use attributional evidence from multiple time points to determine affective reactions and helping intentions for people undergoing foreclosure, as well as ideological differences in these attributional processes. Participants read about people who were undergoing foreclosure for onset and offset controllable or uncontrollable reasons and then answer questions about their perceptions of these targets. The results suggested that both onset

and offset controllable information contributed to the emotional reactions and helping intentions of the participants with the participants experiencing more negative affect and less helping intentions when the target was in a controllable onset or offset situation. Conservatives primarily relied on onset controllability information to decide who should receive government aid, while liberals updated their initial attributions with offset controllability information.

**Data availability:** The study is publicly available at: https://www.tessexperiments.org/study/brandt708

**Data pre-processing:** The primary outcome variable is `Q7`. The treatment condition is defined as `XTESS003 = 1` (family can afford the mortgage), and the control condition is defined as `XTESS003 = 2` (family might not afford the mortgage). The following variables are included as covariates: `XPARTY7, XREL1, XREL2, PPAGE, PPGENDER`. The final processed dataset contains $n = 624$ observations.

**Prompting details:** An example prompt is provided below.

---

### Example Prompt

**System Prompt:**

> You are a 75-year-old, not so strong Democrat, gender Female. Additionally, regarding religion you are a Muslim and you once a week attend religious services. You are responding to a survey on perceptions towards people who are facing foreclosure. Your answer must be a single integer without additional text, in JSON format with a key-value pair.

**Treatment condition:**

> Recently the growing number of home foreclosures has put a strain on the financial system, which has weakened the United States economy. Foreclosure occurs when a person is behind on home mortgage payments to their bank and the bank decides to repossess (i.e., take back) the home. People may go into foreclosure for a variety of reasons. We are interested in your perceptions towards people who are facing foreclosure. In the following section you will be presented with a situation that describes some people facing foreclosure. Please carefully read the situation and answer the following questions about your reactions to the situation. Some people have a large monthly mortgage payment because they wanted to purchase a larger house than they needed. Now they are facing foreclosure because they do not want to continue paying the mortgage, even though they are able to afford the payments.

**Control condition:**

> Recently the growing number of home foreclosures has put a strain on the financial system, which has weakened the United States economy. Foreclosure occurs when a person is behind on home mortgage payments to their bank and the bank decides to repossess (i.e., take back) the home. People may go into foreclosure for a variety of reasons. We are interested in your perceptions towards people who are facing foreclosure. In the following section you will be presented with a situation that describes some people facing foreclosure. Please carefully read the situation and answer the following questions about your reactions to the situation. Some people have a large monthly mortgage payment because they wanted to purchase a larger house than they needed. Now they are facing foreclosure because the primary income earner in the household lost their job due to their company closing and they can no longer afford payments.

**Question:**

> Do you strongly oppose or strongly support the following statement: The government should offer help (e.g., time, money, resources, etc.) in an effort to help people in this situation. Reply with an integer from 1 (Strongly Oppose) to 7 (Strongly Support), where 4 is a Neutral stance.

---

### C.2.7 Testing a Theory of Hybrid Femininity [39]

**Abstract:** Although men experience advantages working in highly feminized occupations, they are commonly stigmatized as lesser men by outsiders—the people they meet outside of their occupations—for doing "women's work." This experiment is designed to assess whether a woman who has worked in a hypermasculine occupation would similarly be stigmatized as a lesser woman by workers outside of her hypermasculine occupation, or alternatively, whether she would be viewed more favorably by such outsiders for doing "men's work." Specifically, this study aims to develop and empirically test a theory of hybrid femininity, which specifies the conditions under which hypermasculinity as signaled through occupation creates status and reward distinctions among women in external labor markets. The experiment asks respondents to provide recommended compensation and status ratings for a woman candidate while manipulating the gender-typing of her occupational history as well as her intended target job. By disentangling the underlying mechanisms driving these predicted status and reward differences, this study seeks to shed light on how gender inequality persists, even among women, through the privileging of masculinity over femininity, with important implications for the labor market and society at large.

**Data availability:** The study is publicly available at: https://www.tessexperiments.org/study/melin1066

**Data pre-processing:** The primary outcome variable is Q7_1. The treatment condition is defined as P_41 = 3 (applicant has experience in the Army), and the control condition is defined as P_41 = 6 (applicant has experience in the Cosmetics industry). The following variables are included as covariates: P_IDEO, P_ATTEND, P_RELIG, RELIG, GENDER, AGE, REGION9, RACETHNICITY, INCOME, P_PARTYID. The final processed dataset contains $n = 545$ observations.

**Prompting details:** An example prompt is provided below.

---

**Example Prompt**

**System Prompt:**

> You are a 30-year-old, politically Independent, gender Male, ethnicity Hispanic. Your ideology is slightly liberal. Additionally, your religion is Protestant and you about once a month attend religious services. You live in a state of the Pacific region. Your household has a yearly income of $85,000 to $99,999. This task is part of a larger study on the design of Human Resources (HR) recruiting practices to pre-screen job applicants. Your answer must be a single integer without additional text, in JSON format with a key-value pair.

**Treatment condition:**

> Please imagine you work for a prominent management consulting company. You will be provided with a job description and an applicant's résumé who is applying for a Senior Manager position. After thoroughly reviewing the job description and the applicant's résumé, you will be asked to provide your immediate and uncensored opinion. Job description for your review:
>
> Senior Manager (Consulting) Responsible for: - Leading high performance project teams across the organization - Building professional relationships with key stakeholders - Defining project objectives, roadmaps, and deliverables - Aligning project tactics with project strategy for all new services The successful applicant will be hard-working, results-oriented, and a team player. Required Qualifications: • Bachelor's degree in Business Administration or a related field • 3-5 years of related experience • Comfort with travel regionally or globally (up to 30% of time) • Self-motivated with potential for leadership • Excellent communication skills • Solid computer skills, including Microsoft software products
>
> Applicant's résumé for your review:
>
> Name: Amy Decker Motivated Project Manager with 5 years of experience working in military and defense. Education: Rutgers University (New Brunswick, NJ), May 2017 (Graduated) B.A. in Business Administration,

---

GPA: 3.72/4.00 Work Experience: U.S. Army Project Manager (Active-duty Enlisted), 2014 - Present Fort Dix Military Base (Fort Dix, NJ) - Plan and track progress of entire life-cycle of military and defense projects. - Build and maintain project plans, including actual and forecasted activities and timelines. - Ensure project staffing and timely communications throughout project lifecycle. - Identify and manage project risks. Skills and Interests: Computer: Proficient in Microsoft Office (including Word, Excel, Outlook, and PowerPoint). Interests: Running and traveling.

**Control condition:**

Please imagine you work for a prominent management consulting company. You will be provided with a job description and an applicant's résumé who is applying for a Senior Manager position. After thoroughly reviewing the job description and the applicant's résumé, you will be asked to provide your immediate and uncensored opinion. Job description for your review:

`[Job description, same as above]`

Applicant's résumé for your review:

Name: Amy Decker Motivated Project Manager with 5 years of experience working in military and defense. Education: Rutgers University (New Brunswick, NJ), May 2017 (Graduated) B.A. in Business Administration, GPA: 3.72/4.00 Work Experience Cosmetics Project Manager 2014 - Present Precious Cosmetics (Lodi, NJ) - Plan and track progress of entire life-cycle of cosmetics and beauty product projects. - Build and maintain project plans, including actual and forecasted activities and timelines. - Ensure project staffing and timely communications throughout project lifecycle. - Identify and manage project risks. Skills and Interests: Computer: Proficient in Microsoft Office (including Word, Excel, Outlook, and PowerPoint). Interests: Running and traveling.

**Question:**

On a scale from 1 "Not at all" to 7 "Extremely", to what extent do you perceive this applicant as MASCULINE.

### C.2.8 Understanding White Identity Management in a Changing America [45]

**Abstract:** This paper examines how White Americans manage their identity amidst societal shifts using a new measure of advantaged identity management, representative data (N = 2648), and latent profile analysis. The findings reveal five subgroups of White Americans, each managing their identity differently. Four profiles correspond to the main advantaged identity management strategies (defend, deny, distance, dismantle), with a fifth using strategies flexibly. Of 15 predictions regarding how valuing hierarchy, meritocracy, and egalitarianism predict profile membership, 13 were supported. These profiles show contrasting attitudes toward social change, with defender-deniers opposing, denier-distancers moderately opposing, distancers remaining neutral, and dismantlers supporting change. These findings provide some of the first empirical evidence for a theorized model of white identity management and suggest that how White Americans manage their identity has important implications for social change.

**Data availability:** The study is publicly available at: https://www.tessexperiments.org/study/shuman1643

**Data pre-processing:** The primary outcome variable is Q5D. The treatment condition is defined as RND_01 = 1 (disadvantage black people), and the control condition is defined as RND_01 = 0 (advantage white people). The following variables are included as covariates: AGE, GENDER, RACETHNICITY, EDUC5, REGION9, IDEO, PartyID7, RELIG, ATTEND, INCOME. The final processed dataset contains $n = 1623$ observations.

**Prompting details:** An example prompt is provided below.

**System Prompt:**

> You are a 41-year-old individual with gender Male, ethnicity Asian, and with Bachelor's degree education. You live in a state of the New England region. You hold Moderate views and are not so strong Democrat. Additionally, your religion is Atheist and you attend religious services never. Your household has a yearly income of $175,000 to $199,999.

**Treatment condition:** The general purpose of this study is to examine the attitudes of people regarding social issues in America today. You will now be presented with an infographic:

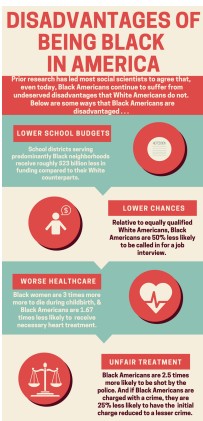

**Control condition:** The general purpose of this study is to examine the attitudes of people regarding social issues in America today. You will now be presented with an infographic:

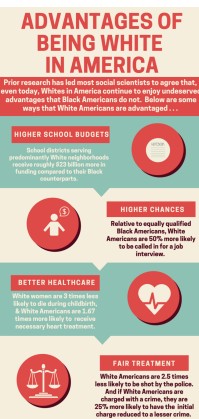

**Question:**

> Rate the extent to which you agree with the following statement from 1 (STRONGLY DISAGREE) to 7 (STRONGLY AGREE): "There should be large scale criminal justice reform to address racial inequalities in the justice system." Your answer must be in JSON format with a single key-value pair.

### C.2.9 Introducing variability in multi-prompt experiments

The user prompts described in the previous section include a final question or instruction sampled from a predefined pool to introduce variability in the multi-prompt settings. Below are some examples of such instructions:

- "Consider all relevant factors and place this on the scale."

- "Reflect on the scenario and use your reasoning to assign a value."
- "From your understanding of the situation, quantify this feeling."
- "Given your insights and the context described, provide your evaluation."
- "With the provided details in mind, rate your feeling on the scale."
- "Consider all the information and your perspective to choose a suitable score."
- "Evaluate the feeling here and align a number with your reasoning."
- "Use the scale provided and your judgment to determine your feeling."
- "Judge this scenario thoughtfully, considering the context and the details shared."
- "Reflect on the key aspects provided and numerically assess your feeling."

