# OpenReview forum: "Efficient Randomized Experiments Using Foundation Models"
_NeurIPS.cc/2025/Conference — NeurIPS 2025 poster_

### Official Review · Reviewer_cefv · 2025-06-14

**Clarity:** 4
**Significance:** 4
**Originality:** 3
**Rating:** 4
**Confidence:** 3

**Summary:**

The paper introduces a novel statistical method H-AIPW estimator, to increase the statistical precision of randomized experiments by leveraging predictions from several foundation models.  The idea from my understand is to get the best convex combination of all the estimators. Theoretically, the paper proves that the H-AIPW estimator is consistent, asymptotically normal, and has an asymptotic variance that is guaranteed to be no larger than the standard estimator that relies solely on experimental data.

**Questions:**

Please see above.

**Ethical Concerns:**

["NO or VERY MINOR ethics concerns only"]

**Final Justification:**

I sincerely thank the authors for the clarification. However, I do not see significant difference. I choose to retain my score.

**Limitations:**

Please see above.

**Quality:**

4

**Strengths And Weaknesses:**

# Strength
1. From my perspective, the problem itself is very important. How to utilize the foundation models in experimentation and causal inference has been of great interest recently.
2. The technical part is quite solid.  The proof that H-AIPW's variance will be no worse than the standard AIPW estimator is interesting.
3. The paper is well written. I enjoy the flow of the paper a lot.

# Weakness
1. My only major concern is that the core methodological idea—combining multiple AIPW estimators via optimal linear weighting to reduce variance—is conceptually close to classical shrinkage estimation, model averaging, and ensemble methods in semiparametric inference. Please do not get me wrong. I like simple ideas, which can usually be more useful in practice.
2. Another minor point is what the unique feature of foundation models is used for the estimator? It seems that any prediction models trained on different dataset (not necessarily foundation model) can be included?

---

> ### Author Rebuttal · Authors · 2025-07-27
>
> Thank you for highlighting the importance and clarity of the paper. We hope this response addresses your questions, but please let us know if you have any remaining concerns that would prevent an “accept score”.
>
>
> **Relation to shrinkage estimators.** We agree that the idea of combining multiple estimators to reduce variance is conceptually close to classical shrinkage estimators. However, we would like to emphasize a key distinction: unlike traditional shrinkage approaches, our method only includes unbiased estimators in the combination. We believe one of the contributions of our work is to show that it is possible to achieve variance reduction without introducing bias. This is a crucial property for practical adoption, especially in high-stakes applications. That said, we appreciate the connection and will expand our discussion of this relationship in the related work section of the camera-ready version.
>
> **Relation to ensemble methods and model averaging**. Ensemble methods such as bagging, random forests, boosting, and the Super Learner have long been used in semiparametric inference, typically combining models trained on the same (often small) dataset. In contrast, our approach integrates predictive models trained on vast amount of *external data, independent of the dataset at hand*. Further, our goal here is not to boost predictive accuracy per se, but to enable valid and more precise inference for treatment effects by leveraging these externally trained models.
>
> **Re foundation models.** Our framework is model-agnostic, and any externally trained prediction model can, in principle, be incorporated - it does not need to be a foundation model. We focus on foundation models in this paper because all our experiments are conducted using LLMs. Moreover, we expect foundation models to be the biggest use case of our general methodology, particularly within the social sciences. We will make this clearer in the camera-ready version to avoid any misunderstanding that our method is limited to foundation models.

---

> > ### Comment · Reviewer_cefv · 2025-08-01
> > **Thank you for your clarification**
> >
> > Thank you for addressing my concerns. I appreciate the effort you have put into your response.
> >
> > Regarding the connection to shrinkage estimators, I understand that you are linearly combining multiple unbiased estimators while maintaining an overall unbiased result. I am trying to grasp the high-level intuition behind how this linear combination can result in a smaller variance than any of the individual estimators. Could you elaborate on whether this is due to any correlation among the estimators?
> >
> > Thank you again for your time and clarification.

---

> ### Author Response · Authors · 2025-08-01
>
> That’s a great question.
>
> **TLDR:** The combined estimator will have strictly lower variance than any of the individual estimators, except for a specific edge case where the correlation among estimators is equal to a critical threshold.
>
> ----
> For simplicity, let's consider two unbiased estimators, and form a linear combination:
>
> $$
> T_\lambda = \lambda T_1 + (1 - \lambda) T_2, \quad  \lambda \in \mathbb R.
> $$
>
> The variance of this combined estimator is:
>
> $$
> \operatorname{Var}(T_\lambda) = \lambda^2 \sigma_1^2 + (1 - \lambda)^2 \sigma_2^2 + 2\lambda(1 - \lambda) \sigma_{12}.
> $$
>
> The weight that minimizes this variance is:
>
> $$
> \lambda^\star = \frac{\sigma_2^2 - \sigma_{12}}{\sigma_1^2 + \sigma_2^2 - 2\sigma_{12}}.
> $$
>
> And the variance of the combined estimator is:
>
> $$
> \operatorname{Var}(T_{\lambda^\star}) = \frac{\sigma_1^2 \sigma_2^2 - \sigma_{12}^2}{\sigma_1^2 + \sigma_2^2 - 2\sigma_{12}}.
> $$
>
> ---
> **Independent estimators**
>
> Now, if $T_1$ and $T_2$ are independent, then $\sigma_{12} = 0$, and the variance simplifies to:
>
> $$
> \operatorname{Var}(T_{\lambda^\star}) = \frac{\sigma_1^2 \sigma_2^2}{\sigma_1^2 + \sigma_2^2}
> $$
>
> This value is always strictly less than both $\sigma_1^2$ and $\sigma_2^2$, meaning that the combined estimator improves on both $T_1$ and $T_2$.
>
> ---
>
> **Correlated estimators**
>
> For correlated estimators, the improvement depends on the degree of correlation.
>
> Assume without loss of generality that $\sigma_1^2 \le \sigma_2^2$, i.e. $T_1$ has the lower variance.
>
> Define the correlation coefficient:
>
> $$
> \rho = \frac{\sigma_{12}}{\sigma_1 \sigma_2}.
> $$
>
> Then the difference between the smaller individual variance and the combined variance is:
>
> $$
> \sigma_1^{2} - \operatorname{Var}(T_{\lambda^\star})
> = \frac{\sigma_1^2 \left(\sigma_1 - \rho \sigma_2\right)^2}{\sigma_1^2 + \sigma_2^2 - 2\rho \sigma_1 \sigma_2}.
> $$
>
> Assuming the two estimators are not perfectly correlated, the expression above is always non-negative, and is strictly positive unless:
> $
> \rho = \frac{\sigma_1}{\sigma_2}.
> $
>
> The intuition is that for this specific correlation level $\operatorname{Cov}\bigl(T_1,T_1-T_2\bigr)=0$, and                                                the difference $T_1-T_2$ is orthogonal to $T_1$ (carries no information about $T_1$’s random error).
>
> Therefore, the combined estimator has smaller variance unless the correlation is equal to the ratio of the standard deviations of the two estimators (or the two estimators are perfectly correlated).

---

> > ### Author Response · Authors · 2025-08-05
> >
> > Thank you for the time dedicated to your initial review and discussion.
> >
> > We believe we have thoroughly addressed the concerns you raised and appreciate the opportunity to clarify our work.
> >
> > As the discussion period is drawing to a close, we wanted to follow up and hear your thoughts on our response. Is there any specific concern you think remains unaddressed?

---

> > > ### Comment · Reviewer_cefv · 2025-08-05
> > > **Thank you for your clarification.**
> > >
> > > Thank you for the response. That makes sense. I do not have further comments. I appreciate your time! I will finalize my evaluation soon.

---

### Official Review · Reviewer_j1Ah · 2025-06-26

**Clarity:** 2
**Significance:** 3
**Originality:** 2
**Rating:** 3
**Confidence:** 1

**Summary:**

The paper introduces H-AIPW, a novel estimator for potentially improving the efficiency of randomized experiments by using the predictions of multiple foundation models (rather than just one model). The key motivation to use multiple foundation models seems to be i) that multiple models are available, ii) an individual model might be biased in its predictions.

Overall, the problem of improving the efficiency of designs is an important problem in many disciplines.

**Questions:**

Why not use n=10, n=20, n=30 rather than n=100 and n=200 especially given the motivation in the introduction? You clearly cite the clinical trials in the introduction yet if you can get to n=200 you clearly have no recruitment problem.

Can you empirically show the advantage of 3 models for table 1? Why do you restrict it to these three and why do not not compare to just having the best model and giving it the same compute as to all three? It might be that only using the best model actually works much better than having multiple models which are than worse by definition than the best one?
This is related to your analysis that "suggesting that its gains over other approaches that integrate external models
are not only due to the access to multiple models."  This seems to be true in general and raises the questions what the whole point of the motivation of using multiple models is? Can you provide more detailed results for the benefits of multiple models vs providing the same amount of compute to the best model?

How do you ensure that your assumptions for Proposition 1 are actually met in practice e.g. if n=10? How do you do sample splitting then?

Can you show the effect of effect of sample splitting in n=10 or even n=50? This is in particular important since one of the raised critisms of PPI is that (from related work):  "This risk of increased variance is a critical limitation in many settings—for example, in clinical trials, pharmaceutical sponsors are highly risk-averse and methods that carry even a small chance of underperforming the established standard face significant barriers to adoption" Yet sample splitting and double ML approach likewise will increase the variance of your estimator.

"This approach has already shown promising results in replicating the results of randomized experiments in several scientific disciplines, including medicine [17] and social sciences (see e.g. [3, 5, 4])" -> [5] is an opinion piece and 3 and 4 are technical reports without any clear evidence of promising results. Where do you see the promise of the approach in social sciences?

"Further, as is often the case with language models, multiple competing models may be available, with no clear way to determine the best choice for a given task in advance" -> Is that not a contradiction to the principles of foundation models?

Rather than variance why not focus on the effective sample size for comparing approaches e.g. in Table 1?

**Ethical Concerns:**

["NO or VERY MINOR ethics concerns only"]

**Final Justification:**

I am unsure about the evaluation and rigorous benchmarking of the approach. I am however not a domain expert and the approach seems theoretical sound.

**Limitations:**

A limitation section is completely missing. "We clearly discuss limitations in the conclusion, particularly the reliance on
underlying foundation models’ accuracy and alignment with the experimental domain." This is one sentence in a 8 line conclusion where two sentence before laud the approach.

**Paper Formatting Concerns:**

A limitation section is completely missing. "We clearly discuss limitations in the conclusion, particularly the reliance on
underlying foundation models’ accuracy and alignment with the experimental domain." This is one sentence in an 8 line conclusion, where two sentence before laud the approach. Given the field of application in sensitive areas, this is simply insufficient.

**Quality:**

2

**Strengths And Weaknesses:**

The paper has interesting applications, but the weaknesses and strengths of the approach are insufficiently explored and ablations are completely missing.

I will provide more detailed feedback and questions in the questions section.

---

> ### Author Rebuttal · Authors · 2025-07-27
>
> We hope this response addresses your questions, but let us know if you have any additional concerns.
>
> **Why not use very small sample sizes in experiments (e.g. n=10,n=20,n=30)?** We appreciate the interest in understanding the limits of our method under small-sample regimes. However, we believe it is important to emphasize that n = 100 represents a far more realistic and practically meaningful setting than n = 10 or n = 20. Well-designed clinical trials rarely operate with fewer than 100 total participants. Even Phase  2 trials typically enroll 130 to 300 participants on average, depending on the disease area (see, e.g., Table 2 in [1]). Our choice of sample sizes in simulation studies is consistent with this real-world setting.
>
> Therefore, we do not believe ablations with n = 10, 20, or 30 would provide useful insight. These sample sizes lie far outside the domain of established practice and produce conditions where even the simplest estimators (e.g., difference in means) fail to achieve nominal coverage due to severe data scarcity. In such regimes, inference becomes fundamentally unreliable - not just for our method, but for the broader class of semiparametric estimators.
>
> Finally, we note that Proposition 1 is built on standard and widely accepted assumptions in the semiparametric literature, adapted to our setting. As discussed above, small sample sizes like n=10 fall well outside the range of well-designed experiments and beyond the scope where any statistical estimator remains reliable.
>
>
> **Where do you see the promise of the approach in social sciences? The papers cited are technical reports without any clear evidence of promising results.** We respectfully disagree. The claim that [3] is a technical report is factually incorrect - the paper is published and has collected over 800 citations. As the authors state: "We suggest that language models with sufficient algorithmic fidelity thus constitute a novel and powerful tool to advance understanding of humans and society across a variety of disciplines." This reflects growing recognition of foundation models as a transformative tool in the social sciences.
>
> Further, we disagree with the claim that [4] offers no clear evidence of promising results. On the contrary, the paper presents one of the most comprehensive and rigorous evaluations to date of large language models' ability to predict the outcomes of social science experiments. They use GPT-4 to forecast 476 treatment effects across 70 nationally representative experiments. Overall, the authors report a correlation of r = 0.85 between GPT-4’s predicted and actual treatment effects.
>
>
> **It might be that using the best model actually works much better than having multiple models?** Our framework is flexible enough that it may ultimately rely on a single best-performing model. However, when different models capture complementary signals, a weighted combination can outperform any individual model, making the combination strictly better in such cases.
>
> Nevertheless, we want to clarify that the best single model cannot outperform the combined estimator in terms of efficiency. This is guaranteed by Theorem 1, which establishes that the hybrid estimator’s variance is no worse than that of any individual component estimator. Thus, combining models is never harmful and often beneficial.
>
> The use of multiple models is a flexible feature of our framework, not a requirement. It can be particularly valuable in high-stakes settings like clinical trials, where pre-specifying the analysis plan is often necessary and post hoc model selection is not allowed (as it can affect the validity of the inference).
>
>
> **Can you provide more detailed results for the benefits of multiple models vs providing the same amount of compute to the best model?** Figure 4(b) shows that performance plateaus after averaging the prediction over 5  different prompts. In Table 1, we use 10 prompts per model, meaning the total compute budget is already well beyond the point where additional sampling would lead to meaningful improvements. Therefore, we believe that reallocating the same compute to a single model - e.g. using 30 prompts - would offer no added benefit.
>
>
> **Effective sample size in Table 1.** We agree that reporting effective sample size could also  be informative. However, they are dual of each other, and hence the relative ranking between methods would remain unchanged.
>
> **Lack of limitations.** We agree that the limitations deserve more thorough discussion. Due to space constraints, we only briefly addressed them in the conclusion. In the camera-ready version, we will use the additional available space to include a dedicated limitations section covering model alignment.
>
> [1] Costs of Drug Development and Research and Development Intensity in the US, 2000-2018. Aylin Sertkaya; Trinidad Beleche; Amber Jessup; et al.
>
> [3] Out of One, Many: Using Language Models to Simulate Human Samples. Argyle et al.
>
> [4] Predicting Results of Social Science Experiments Using Large Language Models. Luke Hewitt, Ashwini Ashokkumar, Isaias Ghezae, Robb Willer

---

> ### Author Response · Authors · 2025-08-05
>
> Thank you for the time dedicated to your initial review.
>
> We’d appreciate your engagement with our rebuttal, especially as the NeurIPS policy requires reviewers to actively participate in the discussion phase **before submitting** a “Mandatory Acknowledgement.”
>
> We believe a discussion is particularly important here because several key points in the review appear to reflect misunderstandings or factual inaccuracies:
>
> - The claim that [3] is merely a technical report is factually incorrect - it is a published paper with over 800 citations.
>
> - The statement that [4] provides no promising evidence does not reflect the content of the paper, which reports a strong correlation between GPT-4’s forecasts and experimental outcomes across 70 social science experiments.
>
> - The suggestion that sample sizes like n = 10 are appropriate benchmarks ignores the context of real randomized trials, which our method is designed to support - even small Phase 2 clinical trials enroll over 130 participants.
>
> - The concern that combining models may underperform compared to selecting the best one overlooks Theorem 1, which formally proves that the combined estimator is at least as efficient as any single component estimator.
>
> Given the above, is there a specific concern you think remains unaddressed?

---

> > ### Comment · Reviewer_j1Ah · 2025-08-06
> >
> > Thanks for your detailed response.
> >
> > - Regarding the experiments, it is typically interesting to explore the limits of an approach i.e.  what happens at 50 participants or when does the approach break down?
> > - For the models it is still not clear to me if the results are adjusted for compute. i.e. some smaller models are cheaper but can be prompted more often for the same amount of compute while larger models are more expensive. What is the guideline for users wrt to a compute budget and which of the results are adjusted for compute? Assume you adjust for the compute what is the strategy for a user?
> > - I have indeed tried to read the technical report [4] in detail and to read [3] out of interest. I am by far no domain expert but I am still not finding the results convincing in these reports. Partially this might have to do that it is unclear at least to me how to clearly evaluate the results in that application domain. Given that this is just mentioned in the introduction I would not consider it an obstacle. I would trust the results significantly more if indeed clear benchmarks with AB tests or some alternative would be available.
> >
> > I would weight the opinions of the other reviewers higher especially since I am not a domain expert and have adjusted my confidence level.

---

> > > ### Author Response · Authors · 2025-08-06
> > >
> > > Thanks for the response, we appreciate it.
> > >
> > > - We agree it is important to explore the limits of our approach. Figure 5 in Appendix B.2 presents results for a single study in  small sample sizes ($n = 50$). As shown, neither our method nor the standard AIPW estimator (the gold standard for RCT analysis) achieves nominal coverage in this regime. This outcome is expected, as statistical inference guarantees are asymptotic and tend to break down in very small samples. To clarify the small sample regime limitations, we will include results for $n = 50$ across all studies in the appendix of the revised version.
> > >
> > > - Our results are not adjusted for compute, as we did not find it to be a limiting factor in our setting - both small and large models are cheap. Even with the largest models, running the full 10-prompt budget used in our experiments costs only a few dollars per study. Moreover, we observed diminishing returns beyond 5 prompts, further reducing the motivation to pursue compute-adjusted strategies in this work. That said, exploring performance under stricter compute constraints remains a valuable direction for future research.
> > >
> > > - Since this point is raised in the introduction and falls outside the main focus of our contributions, we also consider it a minor issue. We leave the evaluation of these works to domain experts in the social sciences.
> > >
> > >
> > > Please let us know if there is anything else you would like us to clarify.

---

### Official Review · Reviewer_jps4 · 2025-07-02

**Clarity:** 3
**Significance:** 2
**Originality:** 3
**Rating:** 4
**Confidence:** 4

**Summary:**

This paper proposes Hybrid Augmented Inverse Probability Weighting (H-AIPW), a new estimator that combines the classic AIPW estimator by an outcome regression fitted on randomized experimental data, with multiple AIPW estimators built from foundation model predictions. The authors prove that H-AIPW is consistent, asymptotically normal, and has no larger variance than the standard AIPW baseline under the minimal assumptions for randomized experiment if the weights are chosen by minimizing the joint asymptotic variance. The authors provide a recipe for applying H-AIPW with large language models including prompt design and cross-fitting, and showed the empirical results across social science studies that H-AIPW improves precision and maintains the validity of inference.

**Questions:**

Please refer to the Weakness section above.

**Ethical Concerns:**

["NO or VERY MINOR ethics concerns only"]

**Final Justification:**

My major concern about this paper is merely the generalization of the proposed model since the response item theory (survey experiments) is a limited scenario. The authors provided numerical results for binary outcome and promised the results for continuous outcome, and I am convinced to correspondingly raise the score.

**Limitations:**

The authors can explain how participant covariates and synthetic responses are safeguarded when applying LLM foundation models.

**Quality:**

2

**Strengths And Weaknesses:**

Strengths
1. The proposed estimator is a natural and reasonable extension of classic AIPW estimators, and it has good theoretical qualities including consistency, asymptotically normality, and no larger variance relative to the baseline. Meanwhile, the valid inference requires no more than usual assumptions for randomized experiments and propensity score.
2. The proposed framework is able to make use of different kinds of LLM foundation models flexibly. The provided prompt examples and cross-fitting procedures are friendly for replication.

Weakness
I want to share some concerns that may largely impact the generalization of the work
1. All experiments are drawn from social science surveys with scale responses. The paper lacks demonstration in clinical trials or industrial AB tests where the responses are not from survey. Besides, the foundation model predictions may carry potential biases and privacy considerations for synthetic data generation are not addressed.
2. Empirical undercoverage arises when many models are combined on very small sample size. The paper needs to discuss the practical guidance on ensemble size versus sample size.

---

> ### Author Rebuttal · Authors · 2025-07-27
>
> Thank you for the constructive feedback. We hope this response addresses your questions, but please let us know if you have any additional concerns.
>
>
> **All experiments are drawn from social science surveys.** Our evaluation focuses on survey
> experiments due to the wide availability of high-quality, publicly accessible randomized datasets. This setting allows rigorous benchmarking across multiple studies. Further, we made the scope of our paper clear early on in the introduction (lines 54 and 55):
>  > While our methodology applies broadly, we focus our empirical results on social science survey experiments, where large language
> models (LLMs) can provide rich predictive signals.
>
> In contrast, clinical trials typically involve protected and proprietary data, making evaluation challenging. Further, the goal in our paper is to introduce a novel estimator with strong theoretical guarantees and demonstrate its promise within a well-controlled experimental domain. While we are actively working on applying our method in clinical trial settings, we consider that a separate, more application-driven effort (outside the scope of this paper and not well-suited to the NeurIPS venue).
>
> **Foundation models predictions may carry biases.** We want to clarify that our method is robust against any potential bias from the foundation models. Intuitively, this follows from the AIPW estimator being unbiased regardless of the choice of outcome regression $f$ (for any fixed function $f$, even if derived from a language model).
>
> For clarity, we demonstrate this below for the treated arm (a symmetric argument holds for the control arm):
>
> $$
> \mathbb{E} \left[ \frac{A}{\pi} (Y - f(X)) + f(X) \right] = \frac{\pi}{\pi} \mathbb{E}[Y \mid A = 1] - \mathbb{E}[f(X)] + \mathbb{E}[f(X)] = \mathbb{E}[Y \mid A = 1],
> $$
> where we use the fact that in randomized experiments $ A \sim \text{Ber}(\pi)$ and$f$ is an arbitrary fixed function.
>
>
>
> **Privacy considerations.** This is an important point, and we appreciate the reviewer raising it. ​​In a practical setting where privacy is important, the foundation models can be open source ones that are run locally (e.g., LLaMA, DeepSeek and Qwen). Nevertheless, we will discuss this issue carefully in the limitations section, emphasizing that rigorous privacy safeguards may be necessary when applying the method in practice, in particular when used in the medical setting where foundation models may not be as easily run locally. Future work could explore injecting calibrated noise into inputs or applying structured anonymization techniques (e.g., coarsening age or income). While these directions fall outside the scope of our current paper, they open up a promising research area at the intersection of privacy and causal inference.
>
>
>
> **Trade-off between sample size and ensemble size.** There is a clear trade-off in selecting the number of models to combine in small-sample regimes. As shown in Appendix B.2, empirical undercoverage can occur when too many models are ensembled on very limited data. This is a finite-sample effect: increasing the number of models in the ensemble increases the flexibility of the combined outcome model, akin to fitting a linear regression with too many covariates. As a result, asymptotic guarantees still hold, but require larger sample sizes to kick in.
>
> Importantly, these coverage issues vanish completely with more realistic sample sizes (e.g., n = 200), where our method consistently achieves nominal coverage. Indeed, most well-designed randomized experiments do not operate in extremely low-sample regimes (e.g., n = 50), which fall outside the regime where statistical inference remains reliable.
>
> We will clarify these points and offer guidance for practitioners in the camera-ready version.

---

> > ### Author Response · Authors · 2025-08-05
> >
> > Thank you for the time dedicated to your initial review.
> >
> > We believe we have thoroughly addressed the concerns you raised and appreciate the opportunity to clarify our work.
> >
> > As the discussion period is drawing to a close, we wanted to follow up and hear your thoughts on our response. Is there any specific concern you think remains unaddressed?
> >
> >
> > The NeurIPS organizers have encouraged authors to start the discussion, and we would be grateful for the opportunity to ensure that all your questions have been fully addressed.

---

> > > ### Comment · Area_Chair_4Gup · 2025-08-06
> > > **Gentle ping.**
> > >
> > > Dear reviewer jps4. Could you please engage in a discussion with the authors?

---

> > ### Comment · Reviewer_jps4 · 2025-08-07
> >
> > I appreciate the authors providing more illustrations to answer my concerns.
> >
> > I have a small follow-up: I understand the focus of the paper is social science survey experiments and also expect the similar conclusions hold ideally for other experiments with continuous response, given the explanation about unbiasedness. But the prompt or actual performance is still worth illustration by numerical results. It is possible to highlight the focus as "item response theory" or "scaled response" in the title?

---

> > > ### Author Response · Authors · 2025-08-07
> > >
> > > Thank you for the suggestion.
> > >
> > > Our method and theoretical results are formulated in general terms and apply to binary, scaled, and continuous outcomes.
> > > To avoid confusion, we will clarify in the abstract and introduction that the empirical validation focuses on scaled responses, while the framework is broadly applicable.
> > >
> > > Moreover, we have provided additional binary outcome simulations in our response to Reviewer X7Ms and will incorporate them into the final revision. We will also add continuous outcome simulations to further demonstrate the generality of our framework.
> > >
> > > Please let us know if you have any remaining concerns that would prevent a positive score.

---

> > > > ### Comment · Reviewer_jps4 · 2025-08-07
> > > >
> > > > Thank you for the response. I'd like to increase my rating correspondingly.

---

### Official Review · Reviewer_oL2X · 2025-07-02

**Clarity:** 1
**Significance:** 2
**Originality:** 3
**Rating:** 4
**Confidence:** 4

**Summary:**

This paper considers the idea of combining LLM output with real experimental data when computing estimates of the ATE. The set-up combines augmented inverse probability weighted estimators from experiment and various LLMs. They are combined using a linear combination, following the logic of [28].

**Questions:**

See Weaknesses

**Ethical Concerns:**

["NO or VERY MINOR ethics concerns only"]

**Final Justification:**

Thank you for your assurances to improve the clarity and writing quality of the manuscript, with a focus on describing the high-level way in which LLMs are used as a top priority in the abstract/intro.

**Limitations:**

Yes

**Quality:**

3

**Strengths And Weaknesses:**

# Strengths
- Improving the quality of estimators, e.g. by reducing their variance, is particularly important for low-data experimental regimes.
- Across a range of tasks, the use of LLMs appears to reduce the estimator variance compared to standard AIPW.
- The proposed method allows one to use LLMs to improve the AIPW estimation on the experimental data, rather than adding artificial data directly into the estimator, meaning that no bias from the LLM is introduced to the actual estimate.
- The authors conduct a study into using more inference time compute and larger LLMs

# Weaknesses
- Clarity. Part of the general approach is was not clear to me initially when reading the paper. The AIPW estimator introduced in Sec 3.1 depends on (a) the observed data tuples $(Y_i, A_i, X_i)$ and (b) the function $h$. When using an LLM, one could simulate entirely new data of the same form as the experimental data in (a), this is what the abstract and intro would imply is happening. On the other hand, one could simply use the LLM to create a better estimator for the $h$ function, and use that to create a slightly different estimate from the same pool of experimental data. I believe this is what is actually happening, otherwise the claim that the estimator can tolerate arbitrary bias in the foundation model would not be believable.
- Furthermore, a key point of the approach as I understand it is that $h$ does not have to be estimated statistically when using LLMs because certain counterfactuals can be computed from the model that cannot be computed in real life. But again, this point is not really made expressly anywhere in the paper.

---

> ### Author Rebuttal · Authors · 2025-07-27
>
> Thank you for your request for clarification.
>
> We agree that the current framing in the introduction and abstract could be misinterpreted. In particular, we do not simulate entirely new data tuples. Instead, for given covariates $X_i$ from the experimental data, we use the LLM to predict the counterfactual outcome, i.e. $f(X_i,1-A_i) $. These predictions are then pluged into $\psi(h)$, see equation between line 115 and 116.
>
>
> We regret that Figure 2 did not clearly convey how the LLMs are used. To clarify, LLMs serve as the functions $f_1,\ldots, f_k$ that appear in Algorithm 1.
>
>
> We will revise the introduction and abstract in the camera-ready version to make this clearer.

---

### Official Review · Reviewer_X7Ms · 2025-07-03

**Clarity:** 3
**Significance:** 3
**Originality:** 3
**Rating:** 5
**Confidence:** 3

**Summary:**

The authors introduce a method to combine randomized experimental design with predicted outcomes from foundation models to improve estimation efficiency. They leverage ideas from causal inference and proposes an Augmented Inverse Propensity Weighting (AIPW) estimator that integrates model-based predictions while retaining unbiasedness from randomization. In the paper, the authors demonstrate that this approach yields lower variance estimates of causal effects than the experimental data, while preserving consistency and asymptotic normality under standard assumptions. To support their metrics, the authors empirically validate the framework on societal experiment data and provide clear prompting processes.

**Questions:**

i) If possible, analyze robustness under misspecified predictive models: how much model error can the AIPW estimator tolerate before its benefits disappear? Consider adding experiments where the model is poor at generalizing to show limits of this method.
ii) Include cost analysis comparing the saved labeling effort vs. the compute cost of using foundation models.
iii) For more clarity, consider adding a schematic diagram illustrating the interaction between: experimental assignment, foundation model prediction, labeled outcomes and final estimator.

**Ethical Concerns:**

["NO or VERY MINOR ethics concerns only"]

**Final Justification:**

I am happy about the authors' response during the rebuttal period. While I am not an expert about randomized experiments in the specific applications of this paper (clinical trials etc.), I lean to accept the paper as it shows clearly how the new estimator improve the experiment evaluation. After reading author's results about testing the robustness and model cost, I am satisfied about their arguments and want to maintain my score. I am not going to increase my rating as it is already not a borderline score and I don't feel confident to increase it.

**Limitations:**

Clearly stated in the paper.

**Paper Formatting Concerns:**

No.

**Quality:**

3

**Strengths And Weaknesses:**

Overally a good paper and an interesting topic.

Strengths:

The paper adapts classical AIPW estimators in a novel way to exploit predicted counterfactuals from foundation models, maintaining consistency while gaining efficiency, and therefore is a practical work. It shows both theoretically and empirically how the estimator reduces variance and performs well even with sparse outcome labels. It also tries to break down sources of variance and bias in the estimator, and the authors cleanly present the advantages of incorporating predictions.

Weaknesses:
i) Model Assumption Dependence: As the authors mentioned, the performance depends on the quality of the predictive model — in the domains where foundation models has weaker performance, discussion about workarounds is preferred. The paper does not formally analyze bias amplification under model errors, and doesn't provide adaptive design in this case.

ii) Although the paper discusses the impact of adding more foundation models, the cost of using large foundation models (e.g., running LLM inference on massive datasets) isn’t clearly addressed.

---

> ### Author Rebuttal · Authors · 2025-07-27
>
> We hope this response addresses your questions, but let us know if you have any additional concerns.
>
> **Model assumption dependence.** First, we want to clarify that our approach does not risk bias amplification under model errors. For example, even if a completely incorrect outcome model is used within the AIPW estimator, the estimator remains unbiased. However, variance reduction depends on model quality - when the predictive model is poor, our method offers reduced benefits. Indeed, we provide a simple result in Appendix A.3 (Lemma 1), showing that the variance depends on the l2-error of the model.
>
> We provide here some additional experiments to understand when the benefits from our method (i.e. reduction in variance) disappear.  In particular, we ablate over the correlation between the external model predictions and the true outcome labels.
>
> We simulated $n=100$ observations with $d=5$ Gaussian covariates $X\sim N(0,I)$. Treatments were assigned at random with probability 0.5. The potential outcome functions are generated via
> $
> g_a(X_i)=  \mathbb{I}  [a + \beta^\top X_i + 0.3 X_{i,0}X_{i,1} + 0.1 X_{i,2}^2 > 0 ].
> $
>
>
> We compute the standard AIPW estimator (using a cross‑fitted logistic regression outcome model on the experimental data) and the H‑AIPW estimator, combining the standard AIPW with the external model obtained by flipping the ground truth outcomes to achieve the desired correlation.
>
> | ρ (correlation) | AIPW variance | H‑AIPW variance |
> | ---------------------- | ---------------------- | --------------- |
> | 0.2                    | 0.00295                | 0.00290         |
> | 0.5                    | 0.00302                | 0.00291         |
> | 0.7                    | 0.00302                | 0.00260         |
> | 0.9                    | 0.00302                | 0.00185         |
>
>
> As expected, when the external model is highly correlated with the outcome, the variance of the H‑AIPW estimator is substantially lower than that of the baseline. For moderate correlation, the variance reduction is modest. When the external model is only weakly correlated, the H‑AIPW variance remains slightly lower than that of the baseline, demonstrating the estimator’s robustness. These results indicate that even with small to moderate correlation, our method continues to offer benefits and, importantly, does no harm.
>
>
> **Cost of using foundation models**. Thank you for raising this point. In our setting, inference costs are modest, as most datasets contain only 100 to 200 datapoints. Specifically, the cost of all the experiments reported in our paper was approximately $200 in API credits. We consider this cost negligible compared to the expenses associated with human experimentation.
>
>
> **Schematic diagram**. We’d be more than happy to adjust Figure 2 in the current manuscript, if the reviewer has suggestions for any particular edits that could clarify the illustration more.

---

> > ### Comment · Reviewer_X7Ms · 2025-08-06
> >
> > Thanks to the authors for trying to address my concerns. I appreciate the authors on their responses especially for the first point with additional simulations on different model correlations. Now the arguments are more clear to me. As I stated in the first review, this paper is technically solid and I will maintain my original rating.

---

### Decision · Program_Chairs · 2025-09-17

**Decision:**

Accept (poster)

**Comment:**

This paper introduces Hybrid Augmented Inverse Probability Weighting (H-AIPW), for improving the statistical efficiency of randomized experiments by integrating predictions from foundation models. It combines the standard, unbiased estimator from experimental data with model-based predictions in a way that reduces variance without introducing bias.

The paper received a largely positive reception from the reviewers. Most reviewers found the work to be technically solid, original, and significant. They highlighted the novelty of adapting classical AIPW estimators to leverage foundation models, the strong theoretical guarantees provided , and the practical importance of improving estimation efficiency, especially in low-data settings.

Initial reviews raised several concerns. However, the authors provided a thorough and convincing rebuttal. They clarified the technical details, presented new simulation results to address robustness and generalization concerns. This proactive engagement satisfied most reviewers, with one raising their score and others maintaining their positive ratings.

One reviewer remains less convinced, questioning the experimental design and the motivation for using multiple models. However, they also state a low confidence in their assessment and recommend weighing the opinions of other reviewers more heavily.

Given the overall enthusiastic response, the technical soundness of the work, and the authors' diligent and effective handling of the reviewers' feedback, I recommend accepting this paper.